# Tracing the electron transport behavior in quantum-dot light-emitting diodes via single photon counting technique

Qiang Su[1,2], Zinan Chen[1] & Shuming Chen [1]✉

The electron injection and transport behavior are of vital importance to the performance of quantum-dot light-emitting diodes. By simultaneously measuring the electroluminescence-photoluminescence of the quantum-dot light-emitting diodes, we identify the presence of leakage electrons which leads to the discrepancy of the electroluminescence and the photoluminescence roll-off. To trace the transport paths of the leakage electrons, a single photon counting technique is developed. This technique enables us to detect the weak photon signals and thus provides a means to visualize the electron transport paths at different voltages. The results show that, the electrons, except those recombining within the quantum-dots, leak to the hole transport layer or recombine at the hole transport layer/quantum-dot interface, thus leading to the reduction of efficiency. By reducing the amount of leakage electrons, quantum-dot light-emitting diode with an internal power conversion efficiency of over 98% can be achieved.

Quantum-dot (QD) light-emitting diodes (QLEDs) are hotly investigated due to their potential applications in low-cost, wide color gamut and flexible displays[1-7]. In the past decade, substantial efforts have been devoted to the material synthesis and device engineering, and breakthrough advances have been achieved that significantly improve the performance of QLEDs[8-17]. Despite the rapid developments, the critical working mechanisms of QLEDs including charge injection/transport[18-23], efficiency roll-off[10,24,25] and device degradation[12,17,19,26-29] are still unclear, limiting the further improvement of device performance. As an optoelectronic device, QLED consumes electrons and converts them into photons. If all injected electrons are fully converted into photons, a quantum conversion efficiency of 100% can be obtained. However, not all electrons are confined within the QDs and subsequently converted into photons, especially when the devices are operated at a small current density ($J$) or a large $J$. Take the external quantum efficiency–current density (EQE-$J$) characteristics (Supplementary Fig. 3) for example, at a low $J$ of 0.06–0.14 mA cm$^{-2}$, although a considerable amount of electrons are injected into QDs, only a small fraction of them are converted into photons, leading to a very low EQE of only 0.32–5.81%. The EQE is gradually climbed up as the $J$ is

increased, and at a high $J$ of 20.36 mA cm$^{-2}$, a maximum EQE of 20.60% is achieved, suggesting that most of the injected electrons are efficiently converted into photons. The maximum EQE can be retained at a certain $J$ level (20.36–135 mA cm$^{-2}$). However, at a large $J$ (>200 mA cm$^{-2}$), the EQE is rapidly decreased, a phenomenon known as efficiency droop or roll-off[10,24,25]. The variation of EQE indicates that the electron transport and recombination behaviors are dynamically changed as the $J$ is increased. Apparently, at both a small $J$ and a large $J$, the injected electrons are not fully converted into photons. If the electrons are not converted into photons, where do they go?

Previous studies have revealed that, at a large $J$, Auger recombination is actively triggered[24,25,30]. This process involves the non-radiative transfer of energy from excitons to nearby electrons, and as a result, the electrons are excited to the higher energy states. In addition, at a large $J$, the electrons that accumulate at QD layer can overflow to the hole transport layer (HTL)[16,19,20], resulting in the formation of leakage current. The hot electrons generated by Auger recombination and the leakage electrons do not convert into photons. Instead, they generate Joule heat, which reduces the quantum yield (QY) of QDs by creating more defects in the QDs or thermally

[1]Department of Electrical and Electronic Engineering, Southern University of Science and Technology, Shenzhen, PR China. [2]School of Physical Sciences, Great Bay University, Dongguan, PR China. ✉e-mail: chen.sm@sustech.edu.cn

dissociating the excitons[10,31]. The presence of Auger recombination, leakage current and Joule heat reduces the number of electrons that are converted into photons, and thus leads to the efficiency roll-off at a large $J$. However, it is still unclear which factor plays the dominant role in causing this phenomenon. In addition, even at a low $J$, where the effects of Auger recombination and Joule heat can be neglected, the device still exhibits a very low EQE, and the underlying causes are not yet fully understood.

What happens to the electrons that do not convert into photons? To answer this fundamental question, a clear physical picture of electron injection, transport and recombination within the devices should be established. Addressing this fundamental question will shed light on the behavior of these non-radiative electrons (referred to as leakage electrons hereafter) and contribute to a comprehensive understanding of the device's operation. Although the behavior of leakage electrons could provide valuable insights, tracking their transport is exceedingly challenging. This difficulty stems from the fact that the leakage electrons produce weak photon signals that are currently undetectable by available instruments and methods. In this contribution, to accurately trace the electron transport paths, a characterization methodology based on the single photon counting (SPC) and the electroluminescence–photoluminescence (EL–PL) co-measurement techniques is developed. The results of the EL–PL co-measurement reveal that the low EQE is primarily caused by leakage electrons, rather than Auger recombination or Joule heat as commonly believed. The leakage paths of the electrons are further probed by using SPC technique, which is capable of detecting the weak photon signals generated by the leakage electrons and thus allows us to depict the electron transport paths at different voltages. The results demonstrate that even at a small $J$, electrons can leak to the HTL and/or recombine at the HTL/QD interface, resulting in a low EQE. At a large $J$, the Auger recombination and Joule heat are mainly responsible for the efficiency roll-off. By reducing the amount of leakage electrons, all injected electrons could be converted into photons, and as a result, an internal power conversion efficiency (IPCE) of over 98% can be realized at an applicable brightness of 718 cd m$^{-2}$. The unambiguous revelations of electron transport behavior not only enhance our understanding of the working mechanism of QLEDs but also provide fresh insights into the development of efficient and stable QLEDs.

## Results and discussion

### Electroluminescence–photoluminescence co-measurement

The QLEDs usually exhibit a positive aging effect (Supplementary Fig. 4), and to exclude the effect of positive aging on device characteristics, all devices were encapsulated and shelf-stored in N$_2$ glove box for several days, so that the positive aging process is fully completed. Figure 1a shows the current density–voltage–luminance ($J$–$V$–$L$), capacitance–voltage ($C$–$V$) and EQE–$V$ characteristics of a typical red QLED (Supplementary Fig. 5). At a sub-bandgap voltage of 1.60 V, the device exhibits a detectable luminance of 0.27 cd m$^{-2}$, and at the same time, the current and the capacitance are abruptly increased, indicating that a considerable amount of electrons start to inject and accumulate in the QD layer. At a sub-bandgap voltage of 1.60–2.00 V, the electrons are injected into QDs first due to their low injection barrier, while a few lucky holes, with the assistance of thermal energy, can overcome the injection barrier and recombine with the electrons, leading to the sub-bandgap luminance of 0.27–700 cd m$^{-2}$, as we disclosed previously[32]. At a sub-bandgap voltage of 1.60–2.00 V, the energy of the injected electrons (1.60–2.00 eV) is smaller than that of the emitted photons (2.00 eV); therefore, if all injected electrons are converted into photons, an IPCE of over 100% can be achieved. However, the device exhibits a very low EQE of 0.32–16.17%, indicating that most of the injected electrons do not convert into photons. As the voltage is increased, the EQE increases rapidly and reaches a maximum

value of 20.60% at 2.25 V; further increasing the voltage causes the EQE to quickly roll off.

To understand the EQE–$V$ characteristics, the factors that affect the EQE should be identified. For a given device structure, the EQE is determined by both the charge balance factor $\gamma$ and the exciton radiative efficiency $\eta_r$. The presence of excess electrons or leakage current can reduce the $\gamma$, while the presence of Auger recombination and Joule heat can lower the $\eta_r$. To distinguish whether the reduction of EQE is due to a decrease in $\gamma$ or $\eta_r$, we simultaneously measured the EL and the PL of QLED using our home-built system. A major advantage of this method is that we can in-situ access the $\eta_r$ when the QDs are being electrically pumped. A direct-current (DC) voltage was applied to a QLED to initiate the EL emission, and at the same time, the QDs were periodically excited by a 532 nm laser. To avoid triggering the Auger recombination, the laser intensity was intentionally kept low (Supplementary Note 1). The generated EL and PL signals were collected by a Si avalanche photodetector (Si-APD). The weak alternate-current (AC) PL signals were separated and picked up by the lock-in amplifier while the DC EL signals were read out by the oscilloscope directly. The EL intensity is corrected based on the EL signal and the driving current, so that the obtained EL actually reflects the EQE of the devices under electrical and optical pumping. The normalized EL and PL intensity as a function of driving voltage are shown in Fig. 1b. Because the EL is determined by both $\gamma$ and $\eta_r$, while the PL is only affected by $\eta_r$, therefore, we can extract the value of $\gamma$ by dividing the EL by the PL. By substituting the peak EQE of 20.60%, the peak QY $\eta_r = 85\%$ and the outcoupling efficiency (Supplementary Fig. 6) $\eta_c = 25\%$ into equation

$$\gamma(V) = \frac{\text{EQE}_{peak} * EL_{normalized}(V)}{\eta_c * \eta_r * PL_{normalized}(V)} \quad (1)$$

the exact $\gamma$ as a function of voltage can be obtained (Supplementary Note 2), as shown in Fig. 1b. It is observed that at a sub-bandgap voltage of 1.60–2.00 V, although the QDs exhibits a high PL intensity, the EL intensity is relatively low (Fig. 1b). There exists a large gap between the PL and the EL (Fig. 1c); such a discrepancy suggests that the low EQE at sub-bandgap voltages is mainly caused by the poor $\gamma$. Further increasing the voltage, the $\gamma$ is rapidly increased, and maximum $\gamma$ of 0.97 is achieved at 2.30–2.50 V, which remains almost unchanged at a voltage of 2.50–5.00 V. At a large voltage of over 6.00 V (corresponding to a high $J$ of 993 mA cm$^{-2}$, see Fig. 1d and Supplementary Fig. 7), the EL decreases more rapidly than the PL, suggesting a decrease in $\gamma$ with increasing voltage. The schematic configuration of the EL–PL co-measurement system is shown in Fig. 1e. And as shown in Fig. 1f, the PL excitation region is entirely located within the uniform EL region, demonstrating that the QDs being evaluated are being electrically and optically pumped.

Through the comparison of EL and PL, it is identified that the low EQE at sub-bandgap voltage (1.60–2.00 V) is mainly caused by the poor $\gamma$. At large voltage (over 6.00 V), although the EQE roll-off is primarily caused by the reduced $\eta_r$, the $\gamma$ also is decreased. The poor $\gamma$ and reduced $\eta_r$ indicate the presence of leakage electrons, which do not contribute to the generation of photons. To improve the device efficiency and stability, it is important to understand the transport/leakage behavior of excess electrons.

### Electron transport paths in QLEDs

The electrons could transport through five possible paths, i.e., path 1: inter-band leakage, paths 2 & 3: direct recombination (including radiative and non-radiative), path 4: overflow leakage and path 5: interfacial recombination, as schematically shown in Fig. 2. The electrons choose to transport through different paths as the driving voltage is varied, as analyzed below.

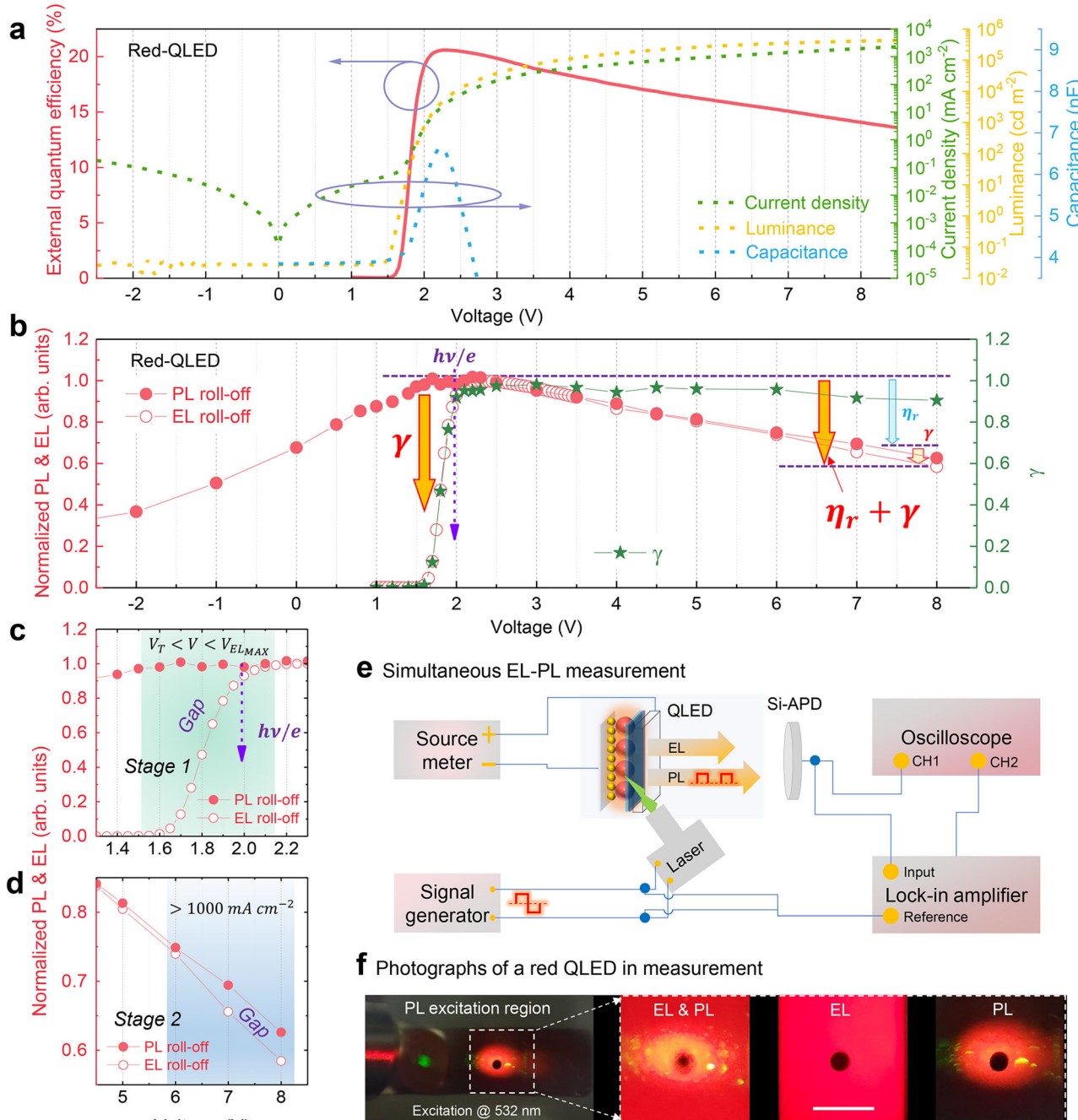

**Fig. 1 | Simultaneous EL–PL measurement of QLEDs. a** The external quantum efficiency–voltage (EQE–$V$), current density ($J$)–$V$, luminance ($L$)–$V$, and capacitance ($C$)–$V$ characteristics of a red QLED. **b** The simultaneous PL–$V$ and EL–$V$ characteristics and extracted $\gamma$ of a red QLED. $\gamma$: charge balance factor, $\eta_r$: exciton radiative efficiency, $h\nu$: photon energy, $e$: elementary charge. The comparison of PL–$V$ and EL–$V$ in **c** *Stage 1* (1.50–2.20 V) and **d** *Stage 2* (>6.00 V) shows two gaps between EL and PL intensity; the gaps are caused by the low $\gamma$. **e** A schematic diagram of the simultaneous EL–PL measurement system. Si-APD: Si avalanche photodetector. **f** The overall photograph, the photograph with both EL and PL on, the photograph with EL on alone, and the photograph with PL on alone in the measurement, scale bar is 1 mm. A monochromatic 532 nm laser was used here, which avoids exciting other functional layers. The laser intensity was intentionally kept low to avoid PL quenching. The photographs were taken through the lens of a PR670 spectrometer. The dark spot is not originated from the QLED; it is an indication of the aperture in the PR670.

(1) $0 < V < V_T$ ($V_T$ defined as the turn-on voltage of the QLEDs). The applied voltage is not high enough to cancel the built-in potential, as shown in Fig. 2a. Due to the presence of built-in potential, it is difficult for both electrons and holes to inject into QDs, and thus there are no detectable photons. At this stage, the electrons can only transport through the inter-bandgap levels (path 1, Fig. 2a) of all functional layers, leading to the generation of ohmic current or inter-band leakage current, as shown in Fig. 2b. The ohmic current is present during the entire operation period, but its impact on device efficiency is negligible, since its value is order of magnitude smaller than that of the recombination current. It is important to note that when the $J$–$V$ curve becomes non-linear in this voltage range, it can no longer be defined as an ohmic current. At this point, variations in this current can significantly affect the efficiency of the QLEDs[29].

(2) $V = V_T$. The built-in potential is completely canceled by the applied voltage, as shown in Fig. 2c. Because the device reaches a

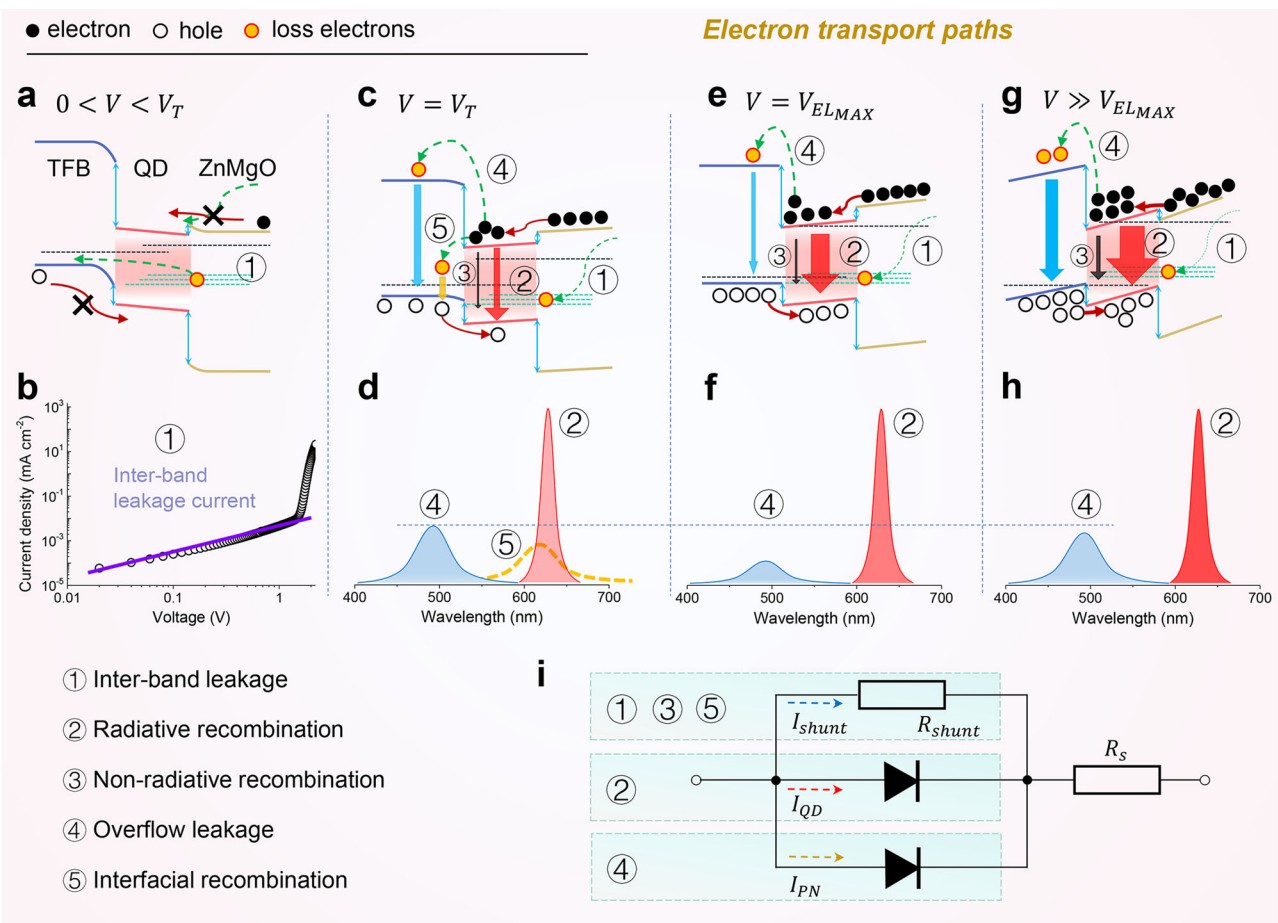

**Fig. 2 | Electron transport paths in QLEDs.** Energy level diagram of a typical red QLED at different voltages. The electrons could transport through five possible paths, i.e., path 1: inter-band leakage, paths 2 & 3: direct recombination (including radiative and non-radiative), path 4: overflow leakage and path 5: interfacial recombination. **a** $0<V<V_T$, the applied voltage is not high enough to cancel the built-in potential, and the electrons can only transport via inter-bandgap levels of functional layers (path 1), leading to the generation of **e** inter-band leakage current. **b** The inter-band leakage current. **c** At $V_T$, flat-band condition is established. Electrons can efficiently inject into QDs while hole injection is enabled by a thermal-assist process. A few electrons recombine with the holes (path 2), while the remaining overflow via path 4 or recombine via path 5. **d** The schematic emission spectra when electrons transport via paths 2, 4 and 5 *at V = $V_T$*. **e** At $V_{EL_{MAX}}$, most electrons recombine via path 2, while those overflow via path 4 are reduced and those recombine via path 5 are eliminated. **f** The schematic emission spectra when electrons transport via paths 2 and 4 *at V = $V_{EL_{MAX}}$*. **g** $V \gg V_{EL_{MAX}}$, most electrons transport through path 2, but not all of them recombine radiatively, and at high voltage, some electrons could leak through the non-radiative path 3 and overflow via path 4. **h** The schematic emission spectra when electrons transport via paths 2 and 4 at $V \gg V_{EL_{MAX}}$. **i** Equivalent circuit model of a QLED.

flat-band condition, a considerable amount of electrons can inject and accumulate in the QDs layer, leading to the abruptly increased luminance, current and capacitance (Fig. 1a). However, due to the presence of a high barrier at TFB/QD interface, only a few energetic holes can inject into QDs via a thermal-assist injection process[32,33]. These lucky holes recombine with a few electrons (path 2, Fig. 2c), leading to the sub-bandgap luminance. The remaining electrons, could either overflow into the TFB (path 4, Fig. 2c), or recombine with the holes accumulated in TFB (path 5, Fig. 2c), consequently resulting in the TFB emission or interfacial emission, as schematically shown in Fig. 2d. These electrons that transport through paths 1, 4 and 5, are referred to as leakage electrons because they do not contribute to QD photons. Due to the presence of leakage paths 1, 4 and 5, the device exhibits a very low EQE.

(3) $V_T < V \leq V_{EL_{MAX}}$ ($V_{EL_{MAX}}$ defined as the voltage corresponding to the peak EQE). With the increasing of voltage, more holes can overcome the injection barrier and inject into QDs, leading to the gradually improved $\gamma$. At $V = V_{EL_{MAX}} = (2.10–2.50)$ V, the applied voltage is larger than the bandgap voltage (1.97 V) of the QDs, and thus a considerable amount of holes could inject into QDs,

resulting in a maximum $\gamma$ and EQE (Fig. 2e). At this stage, the majority of electrons recombine via path 2, while those overflow via path 4 are reduced and those recombine via path 5 are eliminated, as schematically shown in Fig. 2f.

(4) $V > V_{EL_{MAX}}$. The high $\gamma$ can be retained in a wide voltage range (Fig. 2g), indicating that the EQE roll-off at high voltage is primarily caused by the reduced $\eta_r$ due to Auger recombination and Joule heat. At this stage, most electrons transport through path 2, but not all of them recombine radiatively. A part of them leak through the non-radiative path 3. On the other hand, at a high voltage, the $\gamma$ is slightly reduced to 0.90, suggesting that some electrons could overflow via path 4, as shown in Fig. 2h.

Different electron transport paths contribute to different current mechanisms, which can be further analyzed using an equivalent circuit model[8,34,35]. Figure 2i illustrates three distinct current mechanisms: (1) the recombination current through the QDs ($I_{QD}$) resulting from electron transport through path 2; (2) the shunt current ($I_{shunt}$) caused by electron transport through inter-band leakage (path 1), non-radiative recombination (path 3) and interfacial recombination (path 5); (3) the p–n junction current ($I_{PN}$) induced by the direct bypassing of

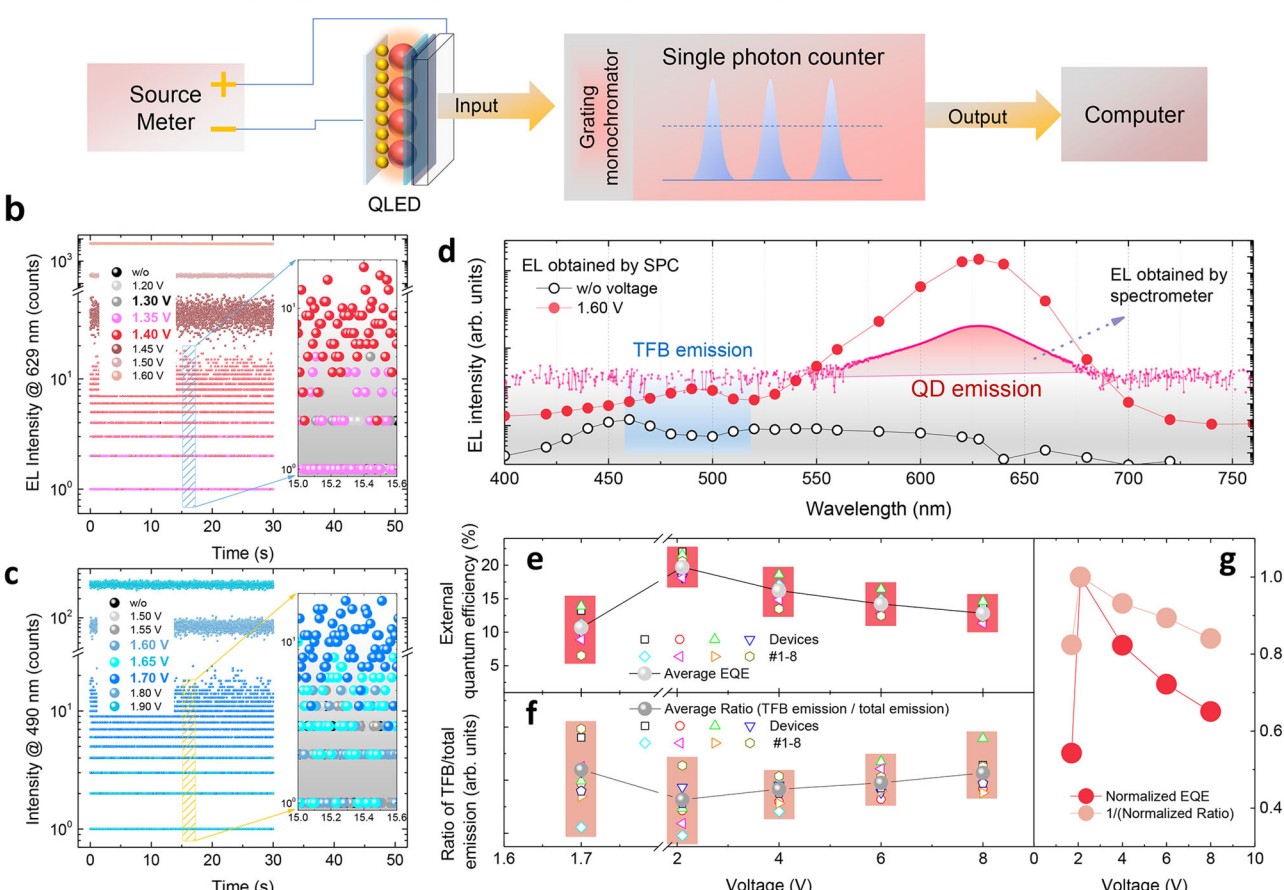

**Fig. 3 | Tracing the leakage electrons in red QLED via single photon counting (SPC) technique. a** The schematic diagram of the SPC measurement setup. The number of photons (EL intensities) at a wavelength of **b** 629 nm and **c** 490 nm, respectively. **d** The EL spectra obtained by SPC and conventional spectrometer.

**e** The EQEs, **f** the averaged ratios of TFB emission relative to total (TFB + QD) emission of eight red QLEDs at different voltage, **g** and the ratio of (TFB emission)/(total emission) is reversely correlated with the EQE.

electrons without recombination in the emissive layer (path 4). The maximization of $I_{QD}$ is the primary objective in devices design. However, $I_{shunt}$ and $I_{PN}$ are present throughout the entire operating voltage range of QLED, significantly affecting QLED's EQE in both small $J$ and high $J$.

**Tracing of the leakage electrons**

If electrons do leak via paths 3 and 4, a fluorescent emission originated from TFB or interfacial charge transfer could be observed. Previous reports indicate that at high voltage, the TFB emission can be detected, suggesting that electrons indeed can overflow via path 4[16,19,20]. However, it remains unclear whether electrons can leak via paths 4 and 5 at low voltage. This is because at low voltage, the fluorescent emission produced by the leakage electrons is so weak that it cannot be detected by conventional spectrometers. To detect such a weak signal, we developed a SPC technique (for further details, see Supplementary Note 3). By counting the number of photons in a given time using a single photon detector, the SPC technique is capable of detecting the weak photon signals. The schematic measurement setup is shown in Fig. 3a. By combing with a grating monochromator, the photons at a specific wavelength can be counted and then recorded by the computer.

We used the SPC technique to detect the photons (wavelength 629 nm) emitted by a red QLED under sub-bandgap bias. As shown in Fig. 3b, the number of photons is gradually increased by increasing the voltage. At a low voltage of 1.35 V, the photons can be clearly detected,

indicating that the $V_T$ of a red QLED can be down to 1.35 V. Compared with the value of 1.60 V measured by a photodetector (Fig. 1a), the detecting threshold is reduced by 0.25 V, indicating the high sensitivity of the SPC technique. Due to the high sensitivity of the SPC, the TFB emission (photons at 490 nm wavelength. For more information see Supplementary Figs. 8, 9 and 10) can be easily detected even at a low voltage of 1.60 V, as shown in Fig. 3c. The results indicate that the leakage path 4 is opened up as soon as the QLED is turned on.

By measuring the number of photons at different wavelength, we are able to plot the full spectra of the devices. As shown in Fig. 3d, at a sub-bandgap voltage of 1.60 V, the emission spectra of QDs obtained by SPC are quite similar with those obtained by conventional spectrometers. However, the spectra obtained by SPC exhibit much improved contrast, while those obtained by spectrometers show high background noises. Due to the high noises, the spectrometers cannot detect weak signals like the TFB emission, because its intensity is below that of the background signal (Supplementary Fig. 11). In contrast, the TFB emission at 1.60 V can be easily discovered by using SPC, which is evident by comparing the spectra obtained at 1.60 and 0 V (Fig. 3c, d). It should be noted that both QD emission and TFB emission are simultaneously observed in the entire voltage range of 1.60−8.00 V for red, green and blue devices (Supplementary Figs. 12 and 13), indicating that electrons not only recombine through path 2, but also overflow via path 4. To evaluate how many percentages of electrons overflow through path 4, the ratio of TFB emission to the total emission (TFB and QD emission) is used (Fig. 3e, f). A higher ratio indicates more

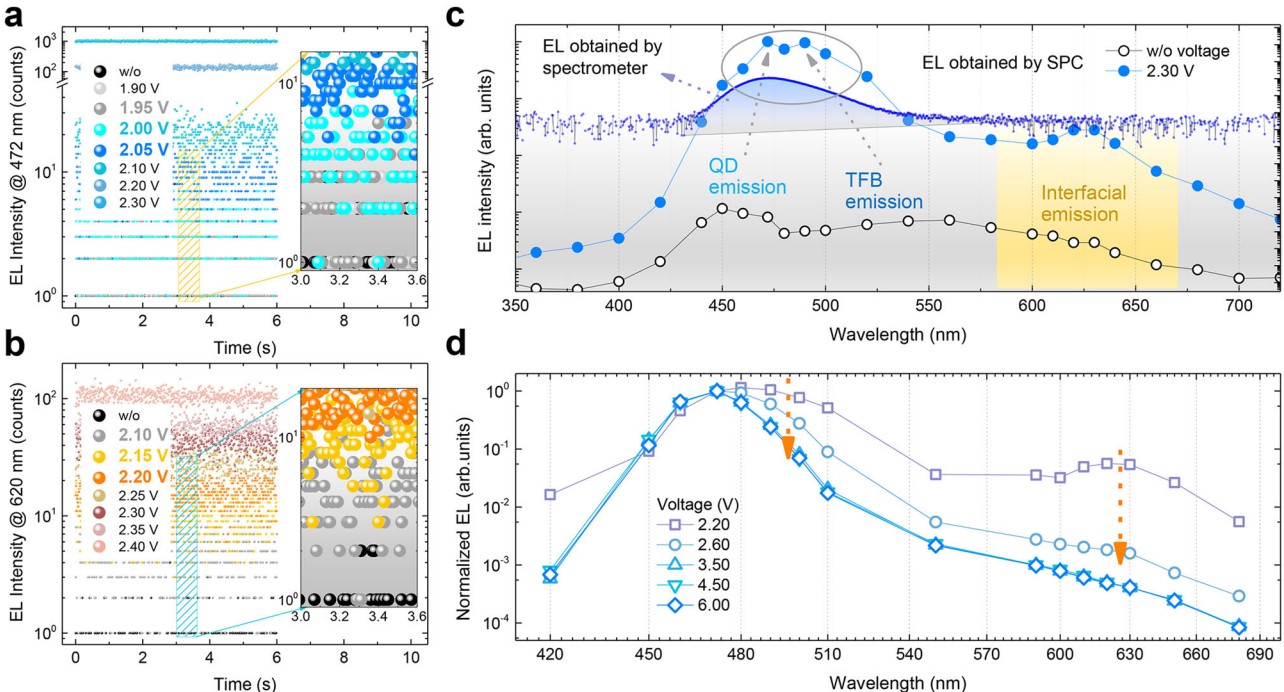

**Fig. 4 | Tracing the leakage electrons in blue QLED.** The photon counts at a wavelength of **a** 472 nm and **b** 620 nm from a blue QLED. **c** The EL spectra obtained by SPC and spectrometer. With SPC, the broad interfacial emission and TFB emission can be clearly observed. **d** The EL spectra obtained by SPC at different voltage. The interfacial emission and TFB emission are gradually reduced as the voltage is increased, and at 3.50 V (corresponding to the voltage of peak EQE), both of them almost disappear.

fraction of electrons overflow via path 4. The ratios at different voltages are compared in Fig. 3f. When the device is just turned on, the ratio is the highest, suggesting that a considerable amount of electrons overflow via path 4. Such a result is reasonable, because at 1.60 V, the $\gamma$ is so low (Fig. 2b) that most electrons cannot find a hole to pair with within the QDs, and eventually they have to overflow via path 4. By increasing the voltage to 2.20 V, the ratio is the smallest, indicating that the fraction of electrons overflow to TFB is greatly reduced, which is in good agreement with the maximum $\gamma$. Further increasing the voltage, the ratio is slightly increased. Apparently, the ratio is reversely correlated with the $\gamma$ (or EQE), as shown in Fig. 3g. With the SPC technique, we are able to trace the leakage paths of the electrons in the entire voltage range, and we disclose that the electrons can always overflow into TFB. The lower the $\gamma$, the higher the overflow could be, which has not been observed before.

Besides transport through paths 1, 2, 3 and 4 as observed in red QLEDs, the electrons could also transport via path 5 by recombining with the holes accumulated at the TFB/QD interface (Fig. 2b). Although path 5 was not detected in red QLEDs, it is easily observed in blue QLEDs by using SPC technique. Figure 4a shows the photon counts at a wavelength of 472 nm (QD emission) of a blue QLED. At a sub-bandgap voltage of 1.95–2.00 V, a considerable amount of blue photons can be detected. Interestingly, we also detected the red photons (wavelength 620 nm) at 2.10 V, as shown in Fig. 4b. Such a red emission is completely different with the intrinsic PL of QDs, indicating the presence of another recombination channel. To disclose its origin, we measured the full spectra of the blue QLEDs using SPC technique. As shown in Fig. 4c, the QD spectra measured by SPC are identical to those measured by spectrometers. However, the spectrometers can only detect the QD emission while other emissions are missed. With the SPC, the hidden information is disclosed. It is revealed that besides the QD emission, there is a TFB emission, and a broad red emission ranging from 580 to 640 nm. Both TFB and red emission are reduced when the driving voltage is increased, as shown in Fig. 4d. The broad red emission is originated from the

interfacial charge transfer[36] (Supplementary Figs. 14 and 15), where electrons in QDs recombine with the holes in TFB. Such an interfacial emission can only occur when there are many carriers accumulated at the interfaces. At a sub-bandgap voltage of 1.95–2.60 V, hole injection into blue QDs is difficult, and thus a large amount of holes have to accumulate at the TFB/QD interface, thereby triggering the interfacial emission. By increasing the voltage, more holes can be injected into QDs, and thus, both interfacial and TFB emission are reduced. At a voltage of 3.50 V, both interfacial and TFB emission almost disappear, and most electrons recombine via path 2, leading to a peak EQE.

Based on the results obtained from the SPC, we now can precisely portray the electrons transport paths within the QLED. Initially ($0<V<V_T$), the electrons transport through the inter-bandgap levels (path 1), leading to the generation of ohmic current that increases linearly. When the devices are just turned on ($V_T \leq V < V_{EL_{MAX}}$), they exhibit very low EQEs due to the poor charge balance. The low $\gamma$ indicates the presence of excess electrons. These excess electrons not only overflow via path 4 (observed in red, green and blue QLEDs), but also recombine via path 5 (observed in blue QLED). As the driving voltage is increased, maximum EQE is achieved; most electrons recombine through path 2 and the fraction of electrons that transport through paths 4 and 5 is the smallest. At a higher voltage, the EQE starts to roll-off, which is mainly due to the Auger recombination and Joule heat that reduces the $\eta_r$. Although a few fraction of electrons still overflow via path 4, most leakage electrons transport through path 3 due to the reduced $\eta_r$.

## Reducing the leakage of electrons

The low $\gamma$ at $V_T \leq V < V_{EL_{MAX}}$ and high voltage (>6.00 V) indicate the presence of excess electrons and thereby the formation of leakage current, which not only reduces the EQE of the devices, but also degrades the device stability. It is reported that excess electrons overflowing into the TFB can cause a structural deformation of TFB and aggravate its stability[16,20]. If the electron leakage paths can be

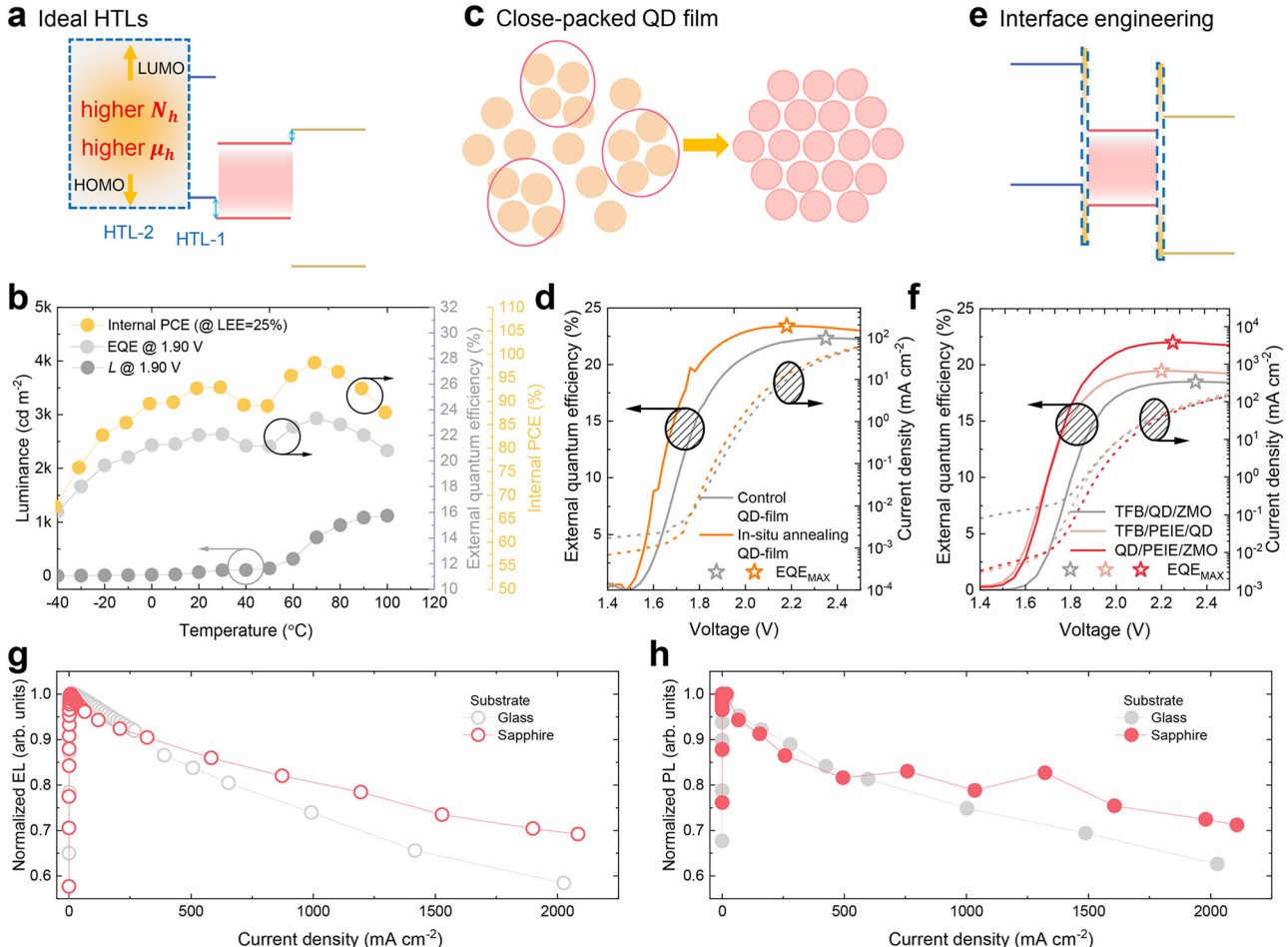

**Fig. 5 | Reducing the leakage of electrons. a** An ideal HTL has a high hole concentration ($N_p$), high hole mobility ($\mu_h$), and better matched energy levels with those of QDs. HOMO: highest occupied molecular orbital level, LUMO: shallow lowest unoccupied molecular orbital level. **b** The $L$–temperature ($T$), EQE–$T$, and IPCE–$T$ characteristics of red QLED (LEE defined as the light outcoupling efficiency of the QLEDs). **c** Close-packed QD film can reduce the leakage of electrons. **d** The EQE–$V$ and $J$–$V$ characteristics of red QLED with control QD film and close-paced QD film. **e** Engineering the interfaces of QDs to block the injection and leakage of electrons. **f** The EQE–$V$ and $J$–$V$ characteristics of red QLED with or without PEIE interface modification layer. The **g** EL–$V$ and **h** PL–$V$ characteristics of red QLEDs with glass or sapphire substrate.

eliminated, then all injected electrons could be used to generate photons, and thus a maximum EQE could be obtained even at a sub-bandgap voltage; in this case, because all electrons are up-converted to high energy photons[32,33,37–39], an IPCE of over 100% could potentially be obtained. Devices with an IPCE over unity are exceptionally efficient and, more importantly, highly stable; this is due to the absence of excess electrons, leakage current, accumulated charges and Joule heat. To reduce the electron leakage, several strategies could be used, as discussed below.

(1) Improving $\gamma$ by enhancing the hole injection. If sufficient holes are injected into QDs, then most electrons can recombine via path 2, and thus the proportion of electrons that leak via paths 4 and 5 is reduced. As schematically illustrated in Fig. 5a, to facilitate the hole injection, one should develop an ideal hole transport materials that process the merits of deep highest occupied molecular orbital (HOMO) level, shallow lowest unoccupied molecular orbital (LUMO) level and high conductivity. However, such a material is currently unavailable. Because at sub-bandgap voltage, hole injection into QDs is assisted by the thermal energy[32], we intentionally increased the temperature of the ambience to enhance the hole injection. Figure 5b shows the luminance, EQE and IPCE as a function of temperature. The red QLED was driven by a sub-bandgap voltage of 1.90 V. By increasing the temperature

from −40 to 30 °C, the luminance and EQE are gradually enhanced, indicating that $\gamma$ is improved. At a higher temperature of 40-50 °C, the EQE is slightly reduced; this is because besides promoting the injection of holes, the ambient heat can also induce emission quenching. At 40-50 °C, the thermal-induced emission quenching may play a dominant role than the thermal-assisted injection, thereby causing the reduction of EQE. When the temperature is elevated to 70 °C, thermal-assisted injection may in turn play a dominant role, thus resulting in a maximum EQE of 23.30% and a luminance of 718 cd m$^{-2}$; by assuming an out-coupling efficiency of 25.00%, the corresponding IPCE can reach 98.23% at an applicable luminance of 718 cd m$^{-2}$.

(2) Improving the compactness of the QD layers (Fig. 5c). Due to their nanoparticle nature, it is difficult to prepare a close-packed QD film. In the case of poor film compactness, there is a possibility that the electrons could directly leak from ZnO to TFB without passing QDs. To obtain a close-packed QD film, the QD solution was pre-heated prior to spin-coating, so as to improve their dispersion. Moreover, an in-situ annealing method was employed during the spin-coating process, whereby the substrate was heated throughout, to achieve better dispersion and prevent agglomeration of the QDs during film formation. By doing so, the compactness of the QD film is greatly improved (Supplementary

Figs. 16 and 17). As a result, devices with in-situ annealing QD film exhibit a higher EQE, and the EQE increases more quickly as voltage increases (Fig. 5d).

(3) Engineering the interface. By inserting a wide bandgap blocking layer PEIE at the interface of ZnO/QD and TFB/QD, the injection of electrons and the leakage of excess electrons can be reduced (Fig. 5e), respectively. Consequently, devices incorporating PEIE demonstrate a higher EQE, and notably, the EQE rises more rapidly with an increase in voltage (Fig. 5f), indicating the reduction of leakage current.

By reducing the leakage current, the EQE at small voltage can be greatly improved, while the EQE roll-off at high voltage can be improved by suppressing the Auger recombination and dissipating the Joule heat. We suggest mitigating the Auger recombination by reducing the density of excitons. This could be achieved through various means such as tailoring the structure of QDs, extending the exciton recombination zone, or accelerating the decay rates of excitons. On the other hand, Joule heat-induced quenching can be effectively addressed through thermal management. For instance, using a sapphire substrate with improved heat dissipation can significantly suppress efficiency roll-off (Fig. 5g, h).

In summary, we address a fundamental question of how electrons transport within the QLEDs. By simultaneously measuring the EL and PL, we reveal the presence of excess electrons that leads to the discrepancy between the EL and the PL. To trace the electron transport paths, a SPC technique is developed, which enables us to detect very weak photon signals and thus allows us to see more information that are usually missed by conventional spectrometers. Based on the SPC results, the electrons transport paths within the QLED can be precisely portrayed. The results indicate that besides recombine within the QDs, the electrons can overflow into TFB once excess electrons are present. At low voltages, due to the poor charge balance, a significant fraction of electrons overflow into TFB, which is responsible for the low EQE of the devices, especially when they are driven by a sub-bandgap voltage. The excess electrons can also leak by recombining with the holes in TFB; however, such an interfacial recombination is only observed in blue QLEDs. By reducing the amount of leakage electrons, all injected electrons could be converted into photons, and as a result, an IPCE of over 98% could be realized at an applicable brightness of 718 cd m$^{-2}$. At high voltages, although electrons overflow into TFB still exist, it is the Auger recombination and Joule heat that mainly responsible for the EQE roll-off. Our unambiguous revelations of electron transport behavior not only enhance our understanding of the working mechanism of QLEDs, but also provide fresh insights into the development of efficient and stable QLEDs.

## Methods

### Materials

Colloidal QDs (red/green/blue) were purchased from Suzhou Xingshuo Nanotech Co., Ltd. ZnMgO nanoparticles in solution were purchased from Guangdong Poly OptoElectronics Co., Ltd. TFB and PF8Cz HTLs were purchased from American Dye Source and Dongguan VOLT-AMP Optoelectronic Technology Co., Ltd., respectively. PEDOT:PSS (CLEVIOS P AI4083) was purchased from Xi'an Polymer Light Technology Corp. The chemicals chlorobenzene and octane were obtained from Aladdin Industrial Corp., while absolute ethanol was obtained from Shanghai LingFeng Chemical Reagent Co., Ltd. The ITO glass with a sheet resistance of 20 Ω sq$^{-1}$ was obtained from Wuhu Jinghui Electronic Technology Co., Ltd.

### Device fabrication

The structure of conventional QLEDs is glass/ITO/PEDOT:PSS (45 nm)/ TFB (40 nm)/QDs (~15 nm)/ZnMgO (40 nm)/Al (100 nm). The structure of EL devices based HTLs is glass/ITO/PEDOT:PSS (45 nm)/TFB @ 40 nm (or PF8Cz @ 40 nm)/ZnMgO (40 nm)/Al (100 nm).

The fabrication process is as follows. First, a 6 min O$_2$ plasma process was performed on the cleaned ITO glass. And the PEDOT:PSS layer was formed by spin-casting its solution at 3000 rpm and baked at 130 °C for 20 min in the atmosphere. Next, the TFB (8 mg mL$^{-1}$ in chlorobenzene) and PF8Cz (8 mg mL$^{-1}$ in chlorobenzene) HTLs were spin-coated at 3000 rpm for 45 s and baked at 130 °C for 20 min in a nitrogen-filled glove box. Subsequently, the QD EMLs (15 mg mL$^{-1}$ in octane for red, 10 mg mL$^{-1}$ in octane for green and blue QDs solution) were spin-coated at 3000 rpm for 45 s and baked at 100 °C for 5 min (the EL devices based HTLs did not require this process). Afterward, the ZnMgO NPs ETLs (20 mg mL$^{-1}$ in ethanol) were spin-coated at 2500 rpm and baked at 100 °C for 10 min. Finally, the Al cathodes were deposited by thermal evaporation in a high-vacuum chamber with a base pressure of 4 × 10$^{-4}$ Pa. In the end, the prepared devices were encapsulated with UV-resin and cover glass in a nitrogen-filled glove box.

Besides, for the fabrication of the close-packed QD film, the QD solution was pre-heated (45 °C) and an in-situ annealing method (keep the substrate at 60 °C throughout the spin-coating process) was employed.

### Characterizations

A Bruker DektakXT stylus profiler was used to characterize the thicknesses of the functional layers. A quartz crystal microbalance was used to monitor the evaporation rates and the thicknesses during the thermal evaporation process. A fiber-optic spectrometer (USB 2000, Ocean Optics) was used to measure the EL spectra of QLEDs and EL devices based HTLs. A performance characterization system consisting of a dual-channel Keithley 2614B programmable source meter and a PIN-25D calibrated silicon photodiode was used to characterize the $J–V–L$ characteristics of QLEDs. An HP4284A LCR analyzer was used to perform $C–V$ test, and the frequency and amplitude of the AC signal are 1000 Hz and 0.05 V, respectively.

The EL–PL co-measurement system was set by using a dual-channel programmable source meter (Keithley 2614B) to drive a QLED, a signal generator (JunCe Instruments, JDS6600) to drive a 532 nm laser (10 kHz), a Si-APD (THORLABS, APD120A2/M) to receive EL emission and PL emission generated by the QLED, a lock-in amplifier (Stanford Research Systems, SR830) to separate and pick up the weak AC PL signals, and a dual-channel oscilloscope (Tektronix, TBS1102) to receive and read out the EL/PL signal.

The leakage electrons were traced by using SPC technique, which is integrated with an Edinburgh FS5 system.

A custom-designed temperature-controllable probe station was used to carry out the temperature-dependent EL spectra and $J–V–L$ characteristics. The temperature was regulated by a temperature control module that can input liquid nitrogen, and the control accuracy of temperature was 0.1 °C.

## Data availability

The data that support the findings of this study are available from the corresponding author upon request.

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

## Acknowledgements

This work was supported by the National Natural Science Foundation of China (Nos. 62174075 and 62404027), Shenzhen Science and Technology Program (Nos. JCYJ20230807093604009, JCYJ20210324105400002 and JCYJ20220530113809022).

## Author contributions

S.C. supervised the work and wrote the final manuscript. Q.S. conducted the experiments, collected the data, drew the figures and wrote the first draft of the manuscript. Z.C. helped to set up the EL–PL measurement system and measured the data. All authors discussed the results and reviewed the manuscript.

## Competing interests

The authors declare no competing interests.
