## [Peer Review File · Nature Communications]

Tracing the electron transport behavior in quantum-dot light-emitting diodes via single photon counting techniqueEditorial Note: Parts of this Peer Review File have been redacted as indicated to remove third-party material where no permission to publish could be obtained.

REVIEWER COMMENTS

Reviewer #1 (Remarks to the Author):

The manuscript reports on the influence of leakage electrons on device performance in QLEDs using an SPC technique. The technique enables to estimate the electron dynamics in QLEDs at various bias conditions, providing new insights. To improve the manuscript, the authors should address the following questions.

1) In the equation (1), two terms, $EL_{normalized}(V)$ and $PL_{normalized}(V)$, are not defined clearly. Based on the experimental data, it can be inferred that each term corresponds to the “peak intensity” of EL and PL emission peaks, respectively. This would not be a problem if EL and PL spectra were perfectly identical. However, as the authors are well aware, EL and PL spectra are not identical—That is, the EL spectrum changes from PL in its peak wavelength and FWHM due to Stoke's shift, and also it contains weak TFB emission, as shown in this study. Therefore, for the accuracy of the calculation, the authors quantify the total emission of EL and PL spectra by integrating the spectra.

2) At high voltages, the authors claimed that low QLED efficiency is due to the “Joule heat” as well as Auger recombination, which are commonly accepted reasons in this field. However, the peak EQE is highest at 70 degrees C, as shown in Fig. 5d. This raises the question of whether Joule heat causes the low performance. Does this mean that the device temperature by Joule heating is higher than 70 degrees C?

3) Is there a reason why the EQE is lower at 40-50 degrees in Fig. 5d than at nearby temperatures?

4) Figures 5e and 5f demonstrate higher efficiency for each strategy at lower voltage regimes. I wonder if these strategies are still valid in the $V > V_{EL_MAX}$ region. It would be informative to provide the EQE plots for the $V > V_{EL_MAX}$ region (or of the entire range) in Supporting Information.

5) To understand the data of the green and blue devices shown in Figure S7, it is necessary to present basic performance data such as I-V-L and efficiency.

Reviewer #2 (Remarks to the Author):

The authors have addressed a crucial issue of QLED performance, efficiency roll-off at low and high current densities, using extensive experimental analysis. The use of single photon counting to trace electron transport paths is innovative and provides new insights into the electron dynamics within QLEDs. The experimental results clearly show the pathways of leakage electrons and their impact on QLED device performances. Based on this finding, the authors suggest possible ways to improve device performances.

Overall, this paper is well-written, and a valuable contribution to understand the QLED operation mechanisms. The reviewer, however, suggests a minor revision to the manuscript. The reviewer's comments and suggested improvements are summarized below:

1. The red emission due to the recombination of electrons with holes accumulated at the TFB/QD interface was observed only in the blue QLEDs. Explain why this interfacial emission is not seen in red and green QLEDs, even though significant potential energy barriers still exist at the interface of TFB with red or green QDs. Could it be that the blue QD layer is too thin or non-uniform, allowing electrons from the ZnO layer to tunnel more easily in blue QLEDs? In this regard, compare the morphology of the red and green QD layers with that of the blue QD layers to see if there are any differences.

2. It is expected that “positive aging effect” can be expected to occur since the EL devices were encapsulated with UV-resin and cover glass as described in the Device fabrication

section in p. 19. Explain whether “positive aging effect” occurs and, if it occurs, address that effect on the experiment results.

3. The authors mention CBP and MoO₃ in p. 20, but there are no experimental results using such materials.

4. Provide additional details on the calibration process for the single photon counting technique to ensure reproducibility.

5. The text contains some typos and grammatical errors. Correct them throughout the manuscript.

In p.4, line 5, ... can be realized at a applicable ...  ... can be realized at an applicable ...

In p.6, Fig. 1a, Current density (ma cm⁻²)  Current density (mA cm⁻²)

In p.10, line 14, By combing with  By combining with

In p.13, line 14, thereby trigging  thereby triggering

In p.18, last line, Absolute ethanol were  Absolute ethanol was

In p.19, line 1, ITO glass were  ITO glass was

Reviewer #3 (Remarks to the Author):

The manuscript by Su et al. introduces a novel technique for tracing electron transport paths by detecting extremely weak leakage electron signals. By minimizing leakage electrons, the authors demonstrate that quantum-dot (QD) light-emitting diodes (QLEDs) with an internal power conversion efficiency of 98% can be achieved. This method is well designed and this work is very interesting. This paper could be published after some revisions.

1.The authors claim that electrons will leak to the hole transport layer (HTL) or recombine at the HTL/QD interface in addition to recombination in quantum dots. Is this conclusion specific to this device structure, or is it universally applicable?

2.The mechanism behind the low turn-on voltage (1.6 V) of the typical red QLED in Figure 1a should be discussed in detail.

3.Further evidence should be provided to visually confirm the improved compactness of the QD layers after thermal treatment.

4.The EL spectra detected by the single photon counting technique of treated red QLED in Figures 5e and 5f should be provided to intuitively demonstrate the reduction of leakage electrons.

5.Do these methods to reduce leakage electrons affect green QLED and blue QLED?

Reviewer #4 (Remarks to the Author):

This manuscript by Su et al. provides valuable insights into the electron transport behavior in QLEDs. However, as shown below, the picture is overly simplified and the conclusions drawn by the authors are often far away from their experimental base. The current version, in its present form, may not meet the publication standards of Nature Communications without resolving these scientific concerns.

Major issues:

1. The EL-PL technique is a routine one and the SPC technique has been reported by the same group earlier (10.1002/adma.202309123). The 'interfacial recombination' idea was proposed by the group in 2018 (10.1002/jsid.681). The parasitic emission from TFB HTLs was documented in literature (10.1038/s41566-022-00999-9). The electron pathway model presented in this manuscript also appeared to align closely with the existing literature (10.1063/5.0185626). Three strategies provided by the authors to reduce electron leakage are not novel and well-recognized within the scientific community. The authors must elucidate the novel aspects of their findings or improvements in techniques over existing literature. Specifically, how does this work differ from the prior studies cited, including their own previous publications?
2. The term 'charge balance factor' requires a clear definition, especially for qualitative discussions. The authors should provide a transparent methodology for determining the exciton radiative efficiency and outcoupling efficiency, which are pivotal for calculating this factor. Without this, the reported value may be misleading.
3. The decrease in PL intensity during the EL-PL experiment may not directly correlate with a decrease in exciton radiative efficiency, as it could also be attributed to reversible charging of the QD film (10.1038/s41467-020-15944-z). The assertion on page 5 needs substantiation or reconsideration.
4. The manuscript presents a counter-intuitive scenario where electron leakage via certain paths decreases with increasing voltage. This requires a more detailed explanation, particularly on pages 8, 13, and in Figure 2g.
5. The attribution of the 500 nm EL peak to TFB emission lacks conviction. Literature suggests that TFB's EL should peak between 425-440 nm, which is in line with the PL and EL provided by the authors in Fig. S4 and 5. Typically, the broad emission spectrum of organic molecules should arise from the same excited state and thus emerge simultaneously. If the emissions at 430-460 nm and 460-500 nm are indeed distinct and separable excited species, then the EL (PL) properties of a TFB film should exhibit variance contingent upon the voltage (energy of the incident photons).
6. The manuscript ascribes the red emission in blue QLEDs to interfacial recombination. It is unclear if this concept extends to red QLEDs. If not, an explanation is needed. If it does, the analysis might be confounded by the omission of interfacial emission analysis. If the radiative interfacial recombination picture is real, the interfacial emission energy should have a direct relationship with the HOMO level of HTL or the LUMO level of QDs. Confusingly, in literature (reported by the authors, 10.1002/jsid.681), when using TFB or PVK as HTL, the 'radiative interfacial recombination' shows up at the same wavelength.
7. The manuscript omits discussion of the ohmic current, which, according to the data (Fig. 5e and f) and literature (10.1007/s12274-021-3354-7 by the authors or 10.1021/acsnano.9b03507 by another group), significantly influences peak efficiency. The authors should address this aspect.

Minor issues:

1. On page 2 of the main text (line 12), and page 4 of the supplementary information (line 4), "no all" should be corrected to "not all".
2. As pointed out by the authors, the turn-on voltage of QLEDs is influenced by the noise-floor of detection system and the device temperature, which thus lacks of a physical entity. As a result, the discussion on different transport paths classified by driven voltage below or above turn-on voltage is ambiguous (Page 7).
3. The origin of the dark spot shown in Figure 1f should be explained to avoid any confusion regarding the experimental setup or results.
4. The manuscript should include details on the calibration process for the single-point detector system used for acquiring broad-band spectra. The detection efficiency of detector

and diffraction efficiency of grating of monochromator may changes significantly against wavelength. The omission of this information is critical as it affects the accuracy and reliability of the data presented.

Overall, this work could become an interesting publication if the authors could clarify the above issues. Alternatively, the authors might eliminate those repetitive studies/analysis and concentrate on the so-called “interfacial recombination” (Point 6 above). To our knowledge, this could be potentially a new and important finding. In this regard, further experimental (such as with different types of quantum dots with variable LUMO levels) and theoretical (such as calculations of possible rates of this unique charge transport channel) studies are expected to solidify their claims.

Reviewer #5 (Remarks to the Author):

🚩 Response to Reviewer #1

General comment: *The manuscript reports on the influence of leakage electrons on device performance in QLEDs using an SPC technique. The technique enables to estimate the electron dynamics in QLEDs at various bias conditions, providing new insights. To improve the manuscript, the authors should address the following questions.*

Our response: Thank you for your efforts in reviewing this manuscript. We sincerely appreciate your positive comments such as “enables to estimate the electron dynamics in QLEDs, providing new insights”, *etc.* Also, your constructive suggestions are very helpful, which greatly help us to improve the quality of this paper.

Comments #1: *In the equation (1), two terms, $EL_{normalized}(V)$ and $PL_{normalized}(V)$, are not defined clearly. Based on the experimental data, it can be inferred that each term corresponds to the “peak intensity” of EL and PL emission peaks, respectively. This would not be a problem if EL and PL spectra were perfectly identical. However, as the authors are well aware, EL and PL spectra are not identical—That is, the EL spectrum changes from PL in its peak wavelength and FWHM due to Stoke's shift, and also it contains weak TFB emission, as shown in this study. Therefore, for the accuracy of the calculation, the authors quantify the total emission of EL and PL spectra by integrating the spectra.*

Response #1: Thank you for your insightful comments. First of all, we extend our apologies for any confuse the equation (1) may have caused you. With regard to the above points, we would like to explain them in detail as follows:

(1) From the EQE definition:

$$EQE(V) = \gamma(V) * \eta_r(V) * \eta_c,$$

$\gamma(V)$ is equal to:

$$\gamma(V) = \frac{EQE(V)}{\eta_c * \eta_r(V)},$$

where EQE and η_r are varied with driving voltage. Thus, it is unreasonable to use a fixed peak EQE and an η_r in above equations because they cannot reflect the change of EQE and η_r of the QDs under different driving voltages. Thus, we simultaneously measured the EL and the PL of QLED using our home-built EL-PL co-measurement system. We made some reasonable replacements in the equation (1): the EQE is replaced by $EQE_{peak} * EL_{normalized}(V)$; η_r is replaced by $\eta_r * PL_{normalized}(V)$, and η_c is a constant. Thus, $\gamma(V)$ is obtained from the below equation [equation (1) in the revised manuscript],

$$\gamma(V) = \frac{EQE_{peak} * EL_{normalized}(V)}{\eta_c * \eta_r * PL_{normalized}(V)}$$

The $EL_{normalized}(V)$, and $PL_{normalized}(V)$ were obtained by measuring the EL-

V and PL-V characteristics simultaneously. The $EL_{normalized}(V)$ is corrected based on the EL signal and the driving current, so that the obtained $EL_{normalized}(V)$ actually reflects the normalized EQE of the devices. The resulting $EQE_{peak} * EL_{normalized}(V)$ are almost the same as EQE (V) of the devices, except that it also reflects the perturbation of PL excitation on QDs.

In addition, because the PL is only affected by η_r , the $\eta_r * PL_{normalized}(V)$ therefore reflects the $\eta_r(V)$.

- (2) Based on the above explanations, we hope to assist you in understanding that EL and PL are not the peak intensities. In fact, they are the function of the driving voltages.
- (3) There is a Stoke's shift and FWHM differences between the EL and PL spectra, but this does not affect the results of this manuscript. As mentioned above, both EL and PL are monitored independently, and are distinguished by DC and AC signals.
- (4) The TFB signal has no effect on the PL and EL measurement results, because a filter which only allows the red emission of QDs to pass was added in front of the detector. In addition, the emission intensity of TFB is so low that can be neglected.

To avoid confusion, a few sentences (marked with yellow color) had been added to the Supplementary Note 2. These are as follows.

(Page 2 in Supplementary Information section):

(2) In order not to collect the excitation and the TFB signals, a long-pass filter was added in front of the detector, so that only the red emission from the QDs can be detected.

(Page 5 in Supplementary Information section):

Supplementary Note 2. The extraction of the γ

From the EQE definition:

$$EQE(V) = \gamma(V) * \eta_r(V) * \eta_c$$

$\gamma(V)$ is equal to:

$$\gamma(V) = \frac{EQE(V)}{\eta_c * \eta_r(V)}$$

The EL and the PL of QLED were simultaneously measured using our home-built EL-PL co-measurement system. We made some reasonable replacements in the above equation: the EQE is replaced with $EQE_{peak} * EL_{normalized}(V)$; η_r is replaced with

$\eta_r * PL_{normalized}(V)$, and η_c is a constant. Thus, $\gamma(V)$ is obtained from the following equation,

$$\gamma(V) = \frac{EQE_{peak} * EL_{normalized}(V)}{\eta_c * \eta_r * PL_{normalized}(V)}$$

The $EL_{normalized}(V)$, and $PL_{normalized}(V)$ were obtained by measuring the EL-V and PL-V characteristics simultaneously. The $EL_{normalized}(V)$ is corrected based on the EL signal and the driving current, so that the obtained $EL_{normalized}(V)$ actually reflects the normalized EQE of the devices. The resulting $EQE_{peak} * EL_{normalized}(V)$ is almost the same as EQE (V) of the devices, except that it also reflects the perturbation of PL excitation on QDs. In addition, because the PL is only affected by η_r , the $\eta_r * PL_{normalized}(V)$ therefore reflects the $\eta_r(V)$.

Comments #2: *At high voltages, the authors claimed that low QLED efficiency is due to the “Joule heat” as well as Auger recombination, which are commonly accepted reasons in this field. However, the peak EQE is highest at 70 degrees C, as shown in Fig. 5d. This raises the question of whether Joule heat causes the low performance. Does this mean that the device temperature by Joule heating is higher than 70 degrees C?*

Response #2: Thank you for your questions.

- (1) Although the ambient heat can induce emission quenching (ref. 1), it can also function positively by promoting the injection of charge carriers, as we demonstrated before (ref. 2). These two effects can simultaneously affect the EQE, causing the EQE behave differently at different temperature. As shown in Figure 5d, when the temperature is elevated from -40 °C to 30 °C, the EQE is gradually increased, which is ascribed to the enhanced charge injection promoted by the ambient heat. At a higher temperature (>70 °C), the EQE is rapidly decreased, which is caused by the thermal-induced emission quenching.
- (2) As shown in Figure 5d, although the maximum EQE is obtained at an ambient temperature of 70 °C, this does not reflect that the device temperature is 70 °C. Considering the inefficient heat conduction from the chamber ambience to the device, **the device temperature should be much lower than 70 °C.**
- (3) **Different with ambient heat, Joule heat is generated by non-radiative recombination current and leakage current, which definitely causes the low performance of the devices.**

References:

- [1] Sun, Y. Z. et al. Investigation on thermally induced efficiency roll-off: toward efficient and

ultrabright quantum-dot light-emitting diodes. *ACS Nano* **13**, 11433–11442 (2019).

- [2] Su, Q. et al. Thermal Assisted Up-Conversion Electroluminescence in Quantum Dot Light Emitting Diodes. *Nature Communications* **13**, 369 (2022).

Comments #3: *Is there a reason why the EQE is lower at 40-50 degrees in Fig. 5d than at nearby temperatures?*

Response #3: Thank you for your question. As mentioned before, although the ambient heat can promote the injection of charge carriers, it can also induce emission quenching. At 40~50°C, the thermal-induced emission quenching may play a dominant role than the thermal-assisted injection, thereby causing the reduction of EQE.

To make the logic of the paper more clear, several sentences were added in page 16 of the revised manuscript, which read as below.

By increasing the temperature from -40 °C to 30 °C, the luminance and EQE are gradually enhanced, indicating that γ is improved. At a higher temperature of 40~50 °C, the EQE is slightly reduced; this is because besides promoting the injection of holes, the ambient heat can also induce emission quenching. At 40~50°C, the thermal-induced emission quenching may play a dominant role than the thermal-assisted injection, thereby causing the reduction of EQE. When the temperature is elevated to 70.00 °C, thermal-assisted injection may in turn play a dominant role, thus resulting in a maximum EQE of 23.30% and a luminance of 718 cd m⁻².

Comments #4: *Figures 5e and 5f demonstrate higher efficiency for each strategy at lower voltage regimes. I wonder if these strategies are still valid in the $V > V_{EL_MAX}$ region. It would be informative to provide the EQE plots for the $V > V_{EL_MAX}$ region (or of the entire range) in Supporting Information.*

Response #4: Thank you for your suggestions. Indeed, these strategies are still valid in the $V > V_{EL_MAX}$ region. Figure R1 shows the EQE plots for the $V > V_{EL_MAX}$ region; as we can see the higher efficiency is still obtained by suppressing the electron leakage.

Figure R1. **a** The EQE-V characteristics of red QLED with control QD film and close-paced QD film. **b** The EQE-V characteristics of red QLED with or without PEIE interface modification layer.

Comments #5: To understand the data of the green and blue devices shown in Figure S7, it is necessary to present basic performance data such as I-V-L and efficiency.

Response #5: Thanks for your suggestions. The J-V-L and EQE-J characteristics of the corresponding green and blue QLEDs are shown in Figure R2.

Figure R2 (reprinted from Supplementary Figure 10 of the revised manuscript). The **ac** J-V-L and **bd** EQE-J characteristics of the corresponding **ab** green and **cd** blue QLEDs in Supplementary Figure 7.

The data have been added in the Supplementary Information section as

Supplementary Figure 10.

Supplementary Figure 10. The **ac** J-V-L and **bd** EQE-J characteristics of the corresponding **ab** green and **cd** blue QLEDs in Supplementary Figure 9.

We hope our responses/revisions satisfactorily address all your concerns. Once again, we thank you for your constructive and helpful suggestions!

Response to Reviewer #2

General comment: *The authors have addressed a crucial issue of QLED performance, efficiency roll-off at low and high current densities, using extensive experimental analysis. The use of single photon counting to trace electron transport paths is innovative and provides new insights into the electron dynamics within QLEDs. The experimental results clearly show the pathways of leakage electrons and their impact on QLED device performances. Based on this finding, the authors suggest possible ways to improve device performances.*

Overall, this paper is well-written, and a valuable contribution to understand the QLED operation mechanisms. The reviewer, however, suggests a minor revision to the manuscript. The reviewer's comments and suggested improvements are summarized below:

Our response: Thank you very much for your efforts in reviewing this manuscript. We sincerely appreciate your positive comments such as “**is innovative and provides new insights into the electron dynamics, well-written, a valuable contribution**”, etc. We appreciate your comments that certainly help to improve the quality of this paper.

Comment #1: *The red emission due to the recombination of electrons with holes accumulated at the TFB/QD interface was observed only in the blue QLEDs. Explain why this interfacial emission is not seen in red and green QLEDs, even though significant potential energy barriers still exist at the interface of TFB with red or green QDs. Could it be that the blue QD layer is too thin or non-uniform, allowing electrons from the ZnO layer to tunnel more easily in blue QLEDs? In this regard, compare the morphology of the red and green QD layers with that of the blue QD layers to see if there are any differences.*

Response #1: Thank you for your insightful comments. As we stated in the manuscript, the interfacial red emission is only observed in the blue QLEDs. This is because the hole injection in blue QLEDs is the most difficult, and as a result, most holes are accumulated at TFB/blue-QDs interface, thereby resulting in a strong interfacial emission. **Compared to the blue QLEDs, the number of hole accumulation in TFB/red or green-QDs is much lower, thus greatly reducing the interfacial emission. Moreover, even if there is a weak interfacial emission, it is difficult to distinguish such a weak interfacial emission from the strong emission of red or green QDs, as its spectra usually overlap with those of red or green QDs.** Due to the weak emission and the overlapped spectra, we were unable to detect the interfacial emission in red or green QLEDs, but there is still a certain probability that it may be present in red or green QLEDs.

Regarding your concern on film morphology, all QD films share the same thickness of around 20 nm. Considering that the dispersity of red/green/blue QDs in solution is close to each other, **the morphology and uniformity of the blue QD film should be similar to that of the red QD film and thus should not be the root cause for the interfacial emission.** Moreover, if the interfacial emission is due to the recombination of electrons directly tunneling from ZnO caused by the thin or non-uniform QD film, its intensity should increase as the driving voltage increases. However, as shown in Figure 4d, the interfacial emission gradually decreases as the driving voltage increases, further confirming that it is related with the accumulation of holes at TFB/QD interface.

Comment #2: *It is expected that “positive aging effect” can be expected to occur since the EL devices were encapsulated with UV-resin and cover glass as described in the Device fabrication section in p. 19. Explain whether “positive aging effect” occurs and, if it occurs, address that effect on the experiment results.*

Response #2: Thanks for your insightful comments. Our devices indeed exhibited the positive aging phenomenon, as shown in Figure R3 (recapped from Supplementary Figure 2 of the revised manuscript). **To exclude the effect of positive aging on device characteristics, the devices were encapsulated and shelf-stored in N₂ glove box for several days, so that the positive aging process is fully completed.** After positive aging, all devices exhibit a stable efficiency and uniform emission, as shown in Figure R4 (recapped from Figure 1 of the revised manuscript), confirming that the positive aging effect has been ruled out.

Figure R3. Positive aging in red QLEDs.

Figure R4. Photographs of a red QLED in measurement.

To clarify this point, a few sentences had been added to the revised manuscript, as follows.

(Page 4 in the revised manuscript):

The QLEDs usually exhibit a positive aging effect (Supplementary Figure 2), and to exclude the effect of positive aging on device characteristics, all devices were encapsulated and shelf-stored in N₂ glove box for several days, so that the positive aging process is fully completed.

And the data have been added in the Supplementary Information.

(Page 8 in the revised Supplementary Information):

Supplementary Figure 2. Positive aging in red QLEDs. **a** Current density-voltage-luminance (J-V-L) and **b** EQE-J characteristics of a red QLED before and after positive aging. To exclude the effect of positive aging on device characteristics, the devices were encapsulated and shelf-stored in N₂ glove box for several days, so that the positive aging process is fully completed. All devices under test exhibit a stable efficiency and uniform emission, as shown in Figure 1f in the main text, confirming that the positive aging effect has been ruled out.

Comment #3: *The authors mention CBP and MoO₃ in p. 20, but there are no experimental results using such materials.*

Response #3: Thank you very much for your careful review. We have carefully checked all errors and made the necessary corrections.

Comment #4: *Provide additional details on the calibration process for the single photon counting technique to ensure reproducibility.*

Response #4: Thank you for your helpful suggestion. To clarify this point, Supplementary Note 3 has been revised as follows.

(Page 6 in the revised Supplementary Information):

Supplementary Note 3. Tracing the leakage electrons in QLEDs via single photon counting (SPC) technique.

The schematic measurement setup is shown in Figure 3a in the main text. The SPC technique utilizes the inherent discrete nature of photon detector output signals under weak light illumination. By employing pulse discrimination and digital counting techniques, it allows for the identification and extraction of extremely faint light signals. In this study, the SPC integrated within the Edinburgh FS5 system was utilized, which is commonly employed for fluorescence lifetime measurements. We employed this system to detect the weak light signals generated within the QLED due to electron leakage. The setup also incorporates a grating monochromator that enables the selective detection of the photons at a specific wavelength.

During the testing process, the QLED sample was placed at the receiving end of the single photon counter, and the testing area was sealed to isolate ambient light. Subsequently, by driving the QLED through a source meter, the number of photons was recorded for each spectrum at different driving voltages.

Other details:

- (1) The grating monochromator integrated in the single photon counter enables the selective detection of the photons at a specific wavelength. The split signal goes directly to the detector without additional amplification.
- (2) The detector in the single photon counter is calibrated with a broad-spectrum standard light source, ensuring an accurate and error-free response at visible wavelength.
- (3) The kinetic scan mode with a corresponding bandwidth was used during the test. The photon counts for each data point were averaged over a collection time of 30 s. During the testing, the emission of the QLED was very stable.
- (4) The ratio in Figure 3e is mainly calculated based on the EL results detected by single photon counter.
- (5) The device position was fixed during testing.

Comment #5: *The text contains some typos and grammatical errors. Correct them throughout the manuscript.*

In p.4, line 5, ... can be realized at a applicable ...  ... can be realized at an applicable ...

In p.6, Fig. 1a, Current density (ma cm-2)  Current density (mA cm-2)

In p.10, line 14, By combing with  By combining with

In p.13, line 14, thereby trigging  thereby triggering

In p.18, last line, Absolute ethanol were  Absolute ethanol was

In p.19, line 1, ITO glass were  ITO glass was

Response #5: Thank you very much for your careful review. We have carefully checked all errors and made the necessary corrections.

We hope our responses/revisions satisfactorily address all your concerns. Once again, we thank you for your constructive and helpful suggestions!

Response to Reviewer #3

General comment: *The manuscript by Su et al. introduces a novel technique for tracing electron transport paths by detecting extremely weak leakage electron signals. By minimizing leakage electrons, the authors demonstrate that quantum-dot (QD) light-emitting diodes (QLEDs) with an internal power conversion efficiency of 98% can be achieved. This method is well designed and this work is very interesting. This paper could be published after some revisions.*

Our response: Thank you for your efforts in reviewing this manuscript. We sincerely appreciate your positive comments such as “**This method is well designed and this work is very interesting**”. Your constructive suggestions are very helpful, which greatly help us to improve the quality of this paper.

Comment #1: *The authors claim that electrons will leak to the hole transport layer (HTL) or recombine at the HTL/QD interface in addition to recombination in quantum dots. Is this conclusion specific to this device structure, or is it universally applicable?*

Response #1: Thank you for your question. Typical QLEDs with colloidal CdSe quantum dots usually use ZnO nanoparticles as the electron transport layer (ETL) material. The use of ZnO ETL enables a low barrier injection and conductive transport of electrons, making the electron injection more efficient than hole. As a result, electrons are usually over injected to the QDs. The over injected electrons accumulate in the QDs, and can overflow to the hole transport layer (HTL), leading to the formation of leakage current that reduces the efficiency and stability of QLEDs. **Electron leakage into HTL is a common issue that has been frequently observed in CdSe-based QLEDs, and is a universal phenomenon for electron-dominant QLEDs, as disclosed by many groups (refs 1, 2).**

Besides leaking into the HTL, some electrons recombine with the holes accumulated at the HTL/QD interface, leading to the formation of interfacial emission. **As we stated in the manuscript, we can only observe the interfacial red emission in the blue QLEDs.** This is because the hole injection in blue QLEDs is the most difficult, and as a results, most holes are accumulated at TFB/blue-QDs interface, thereby resulting in a strong interfacial emission. Compared to the blue QLEDs, the number of hole accumulation in TFB/red or green-QDs is much lower, thus greatly reducing the interfacial emission. Moreover, even if there is a weak interfacial emission, it is difficult to distinguish such a weak interfacial emission from the strong emission of red or green QDs, as its spectra usually overlap with those of red or green QDs. **Although the red interfacial emission is only observed in the blue QLEDs, there is still a certain probability that it may be present in red or green QLEDs.**

References

- [1] Deng, Y. Z. et al. Solution-processed green and blue quantum-dot light-emitting diodes with eliminated charge leakage. *Nat. Photonics* **16**, 505–511 (2022).
- [2] Gao, P. et al. Electron-induced degradation in blue quantum-dot light-emitting diodes. *Advanced Materials* **36**, 2309123 (2024).

Comment #2: *The mechanism behind the low turn-on voltage (1.6 V) of the typical red QLED in Figure 1a should be discussed in detail.*

Response #2: Thank you very much for your suggestion. As shown in Figure 1a, driven by a small voltage of 1.6 V, the red QLEDs can be turned on and emit the photons with a photon energy of 2.0 eV. The turn-on voltage of 1.6 V is smaller than the bandgap voltage of 2 V. This interesting phenomenon is termed “subthreshold turn-on” or “sub-bandgap turn-on”. **The mechanism under the sub-bandgap turn-on has been well disclosed in our early publication [ref. 1]. We revealed that thermal energy play an essential role in the sub-bandgap charge injection processes. At sub-bandgap bias, holes can be successfully injected into QDs via thermal-assisted injection mechanism, thereby enabling the sub-bandgap turn-on and up-conversion EL of the devices.**

To clarify this point, the following sentences have been revised in the revised manuscript. These are as follows.

(Page 4 in the revised manuscript):

At a sub-bandgap voltage of 1.60–2.00 V, the electrons are injected into QDs first due to their low injection barrier, while a few lucky holes, with the assistance of thermal energy, can overcome the injection barrier and recombine with the electrons, leading to the sub-bandgap luminance of 0.27–700 cd m⁻², as we disclosed previously³². At a sub-bandgap voltage of 1.60–2.00 V, the energy of the injected electrons (1.60–2.00 eV) is smaller than that of the emitted photons (2.00 eV); therefore, if all injected electrons are converted into photons, an IPCE of over 100% can be achieved.

Reference:

- [1] Su, Q. & Chen, S. M. Thermal assisted up-conversion electroluminescence in quantum dot light emitting diodes. *Nature Communications* **13**, 369 (2022).

Comment #3: *Further evidence should be provided to visually confirm the improved compactness of the QD layers after thermal treatment.*

Response #3: Thank you very much for your suggestion. As shown in Figure R5 (recapped from Supplementary Figure 12 of the revised manuscript), **the in-situ**

annealed QD film exhibits a stronger PL intensity and absorbance, indicating that more QDs are present in the annealed film. Because the thickness of the in-situ annealed film is similar with the conventional film, the increased QD number therefore implies that the compactness of the film is improved.

We are sorry that at present, we do not have a direct evidence to characterize the compactness of the QD film. We will continue to investigate the preparation methods of high quality QD films in the future and hope to find better characterization schemes as soon as possible and report them in future research results.

Figure R5. **a** PL intensity and **b** absorbance between conventionally spin-coated QD films and in-situ annealing spin-coated QD films

Comment #4: *The EL spectra detected by the single photon counting technique of treated red QLED in Figures 5e and 5f should be provided to intuitively demonstrate the reduction of leakage electrons.*

Response #3: Thank you for your suggestion. Figures 5e and 5f show that the leakage current of devices is effectively reduced, leading to the improved EQE. The reduced leakage current is highly correlated with the reduced emission of hole transport layer (HTL), as disclosed by the single photon counting (SPC) measurements. To prove the above statement, we reduced the leakage current in blue QLEDs by replacing the TFB HTL with PF8Cz HTL. The use of PF8Cz HTL can effectively reduce the leakage current in blue QLEDs due to its low electron affinity and reduced energetic disorder (Figure R6, reprinted from ref. 1), as reported by Deng et. al. [1]. As shown in Figure R7, with PF8Cz, the resulting QLED exhibits a reduced leakage current and higher EQE, a similar result with that in Figures 5e and 5f. Figure R8 (recapped from Supplementary Figure 11 of the revised manuscript) shows the EL spectra obtained by SPC. It can be observed that the device with PF8Cz exhibits a weaker interfacial emission (marked with red circle), which confirms the reduction of electron leakage. We believe that similar SPC results can be obtained for the devices in Figures 5e and 5f.

[Redacted]

Figure R6 (reprinted from ref. 1). Theoretically optimized geometries and reorganization energies (λ) for TFB (left) and PF8Cz (right) dimers. Middle, a comparison of the electronic structures.

Figure R7. The J-V and EQE comparisons of blue QLEDs using TFB and PF8Cz HTL respectively.

Figure R8. The spectra of blue QLEDs were obtained using SPC under different driving voltages.

Reference:

- [1] Deng, Y. Z. et al. Solution-processed green and blue quantum-dot light-emitting diodes with eliminated charge leakage. *Nat. Photonics* **16**, 505–511 (2022).

Comment #5: *Do these methods to reduce leakage electrons affect green QLED and blue QLED?*

Response #5: Thank you for your question. The electron leakage current is fundamentally caused by the unbalance of charge injection as well as the by-pass current. By improving the hole injection, the charge balance in electron-dominant QLEDs can be effectively improved, while by improving the compactness of the QD film, the by-pass current can be greatly reduced. **Therefore, the methods of reducing the leakage current by improving the hole injection or QD compactness are applicable to the electron-dominant green and blue QLEDs.**

We hope our responses/revisions satisfactorily address all your concerns. Once again, we thank you for your constructive and helpful suggestions!

Response to Reviewer #4

General comment: *This manuscript by Su et al. provides valuable insights into the electron transport behavior in QLEDs. However, as shown below, the picture is overly simplified and the conclusions drawn by the authors are often far away from their experimental base. The current version, in its present form, may not meet the publication standards of Nature Communications without resolving these scientific concerns.*

Our response: Thank you very much for your efforts in reviewing this manuscript. We sincerely appreciate your positive comments such as “**provides valuable insights into the electron transport behavior in QLEDs**”, etc. Your constructive comments are very helpful and will certainly help to improve the quality of this paper. We will address your concerns point-by-point in the responses below.

Comment #1: *The EL-PL technique is a routine one and the SPC technique has been reported by the same group earlier (10.1002/adma.202309123). The ‘interfacial recombination’ idea was proposed by the group in 2018 (10.1002/jsid.681). The parasitic emission from TFB HTLs was documented in literature (10.1038/s41566-022-00999-9). The electron pathway model presented in this manuscript also appeared to align closely with the existing literature (10.1063/5.0185626). Three strategies provided by the authors to reduce electron leakage are not novel and well-recognized within the scientific community. The authors must elucidate the novel aspects of their findings or improvements in techniques over existing literature. Specifically, how does this work differ from the prior studies cited, including their own previous publications?*

Response #1: Thank you for your question. We address your concerns point-by-point below.

- (1) First, we agree that the EL-PL co-measurement technique is not invented by us. However, we have **extended its functionality by using this technique to extract γ over the entire driving voltage range, which is novel and has not been demonstrated before.** Second, our previous report (10.1002/adma.202309123) focuses on investigating the degradation of blue QLEDs, and in that paper the SPC technique is only briefly mentioned, whereas **this work specifically focuses on the SPC technique and reports more details on using the SPC technique to trace the electron transport paths, which is a big difference with that of our previous paper.** Actually, the SPC technique was first developed in this work, and then it was used to investigate the degradation of blue QLEDs. Due to the slow review process of this work (it was evaluated by Nature Photonics for almost a year), the paper

(10.1002/adma.202309123) was published before this work.

- (2) As you mentioned, we have previously disclosed the interfacial recombination in QLEDs (10.1002/jsid.681), but at that time we were unable to study the weak interfacial emission at low voltages. In this work, we used the SPC technique to revisit this issue and **revealed a new finding that at low voltages, electrons mainly leak through the interfacial recombination, leading to a weak interfacial emission that can only be detected by the SPC technique.**
- (3) Previous reports only revealed the fluorescence emission due to electron leakage into the TFB in aged QLED (Figure R9, reprinted from ref. 1) or QLED under high driving current (Figure R10, reprinted from ref. 2), whereas in this paper, we **revealed a new finding, enabled by the SPC technique, that even at low voltages near to the turn-on, electrons can still leak into the TFB or recombine interfacially** (Figure R11, recapped from Figure 3 of the revised manuscript), thus providing a correct mechanism for the low EQE at low voltage.
- (4) Regarding your concern about the electron path model in our manuscript, although it is similar to that in the existing literature (10.1063/5.0185626), its correctness has not been verified previously. **With the SPC technique, such a model is experimentally confirmed in this paper.** We have cited this paper (ref. 3) as the ref. 35 in the revised manuscript.

In addition, we would like to further clarify the significance of our work in response to your concerns about the novelty of this study. **The developed SPC technique enables us to detect very weak photon signals and thus allows us to see more information that is usually missed by conventional spectrometers. For example, the electron transport paths at low voltages have not been revealed. With our developed SPC technique, the electron transport and leakage paths at different voltages, ranging from near turn-on to tens of volts, can be precisely revealed, which can enhance our understanding of the working mechanism of QLEDs.**

[Redacted]

Figure R9 (reprinted from ref. 1). Semilog plot (left) and linear plot (right) of EL spectra of a QLED before electrical aging and after aging at 50.0 mA cm^{-2} for 100 h.

[Redacted]

Figure R10 (reprinted from ref. 2). EL spectra of the QD-LEDs (semi-log scale) measured at a current density of 100 mA cm^{-2} or the voltage corresponding to the peak EQE. The parasitic emissions from TFB HTLs are indicated by the arrows.

a Detection of weak light signals with single photon counting (SPC) technique

b

d

c

e

Figure R11 (recapped from Figure 3 of the revised manuscript). Tracing the leakage electrons in red QLED via single photon counting (SPC) technique.

References:

- [1] Ye, Y. X. et al. Design of the hole-injection/hole-transport interfaces for stable quantum-dot light-emitting diodes. *J. Phys. Chem. Lett.* **11**, 4649–4654 (2020).
- [2] Deng, Y. Z. et al. Solution-processed green and blue quantum-dot light-emitting diodes with eliminated charge leakage. *Nat. Photonics* **16**, 505–511 (2022).
- [3] Li, M. et al. The warming-up effects of quantum-dot light emitting diodes: A reversible stability issue related to shell traps. *The Journal of Chemical Physics* **160**, 044704 (2024).

Comment #2: The term 'charge balance factor' requires a clear definition, especially for qualitative discussions. The authors should provide a transparent methodology for determining the exciton radiative efficiency and outcoupling efficiency, which are pivotal for calculating this factor. Without this, the reported value may be misleading.

Response #2: Thank you for your insightful comments. From the mathematical definition of EQE:

$$EQE(V) = \gamma(V) * \eta_r(V) * \eta_c$$

$\gamma(V)$ is equal to:

$$\gamma(V) = \frac{EQE(V)}{\eta_c * \eta_r(V)}$$

where γ is the charge balance factor which characterizes the number of excitons formed per injected charge carrier pair; η_r is the exciton radiative efficiency with a maximum value equal to the PLQY of the QD thin film (85%); η_c is the light outcoupling efficiency. The η_c of our QLEDs is calculated to be 23.84%, as shown in Figure R12. As discussed in our paper, the $\eta_r(V)$ at different voltage can be obtained by in situ measuring the PL-V characteristics when the device is under electrically driving. Because the PL is only affected by η_r , the $\eta_r * PL_{normalized}(V)$ therefore reflects the $\eta_r(V)$. By substituting the $\eta_c=23.84\%$, the measured $EQE(V)$ and the measured $\eta_r(V)$ into above equation, the γ at different voltage can be precisely quantized.

Figure R12 (recapped from Supplementary Figure 4 of the revised manuscript). The

outcoupling efficiency of red QLED with a structure of: glass/ITO (45 nm)/PEDOT:PSS (45 nm)/TFB (40nm)/R-QDs (15 nm)/ZnMgO (40 nm)/Al (100 nm).

To avoid confusion, a few sentences had been added to the Supplementary Note 2. These are as follows.

(Page 5 in Supplementary Information section):

Supplementary Note 2. The extraction of the γ

From the *EQE* definition:

$$EQE(V) = \gamma(V) * \eta_r(V) * \eta_c$$

$\gamma(V)$ is equal to:

$$\gamma(V) = \frac{EQE(V)}{\eta_c * \eta_r(V)}$$

The EL and the PL of QLED were simultaneously measured using our home-built EL-PL co-measurement system. We made some reasonable replacements in the above equation: the *EQE* is replaced with $EQE_{peak} * EL_{normalized}(V)$; η_r is replaced with $\eta_r * PL_{normalized}(V)$, and η_c is a constant. Thus, $\gamma(V)$ is obtained from the following equation,

$$\gamma(V) = \frac{EQE_{peak} * EL_{normalized}(V)}{\eta_c * \eta_r * PL_{normalized}(V)}$$

The $EL_{normalized}(V)$, and $PL_{normalized}(V)$ were obtained by measuring the EL-V and PL-V characteristics simultaneously. The $EL_{normalized}(V)$ is corrected based on the EL signal and the driving current, so that the obtained $EL_{normalized}(V)$ actually reflects the normalized EQE of the devices. The resulting $EQE_{peak} * EL_{normalized}(V)$ is almost the same as EQE (V) of the devices, except that it also reflects the perturbation of PL excitation on QDs. In addition, because the PL is only affected by η_r , the $\eta_r * PL_{normalized}(V)$ therefore reflects the $\eta_r(V)$.

And the simulation results of light outcoupling have been added as Supplementary Figure 4 to the revised Supplementary Information.

Supplementary Figure 4. Light outcoupling efficiency. The outcoupling efficiency (OCE) of a red QLED with a structure of: glass/ITO (45 nm)/PEDOT:PSS (45 nm)/TFB (40nm)/R-QDs (15 nm)/ZnMgO (40 nm)/Al (100 nm). It can be seen that the OCE in red QLED is 23.84%. Therefore, we assume that 25% of OCE is reasonable. Optical properties of the functional layer are referenced from our previous

report [3].

Comment #3: *The decrease in PL intensity during the EL-PL experiment may not directly correlate with a decrease in exciton radiative efficiency, as it could also be attributed to reversible charging of the QD film (10.1038/s41467-020-15944-z). The assertion on page 5 needs substantiation or reconsideration.*

Response #3: Thank you for your insightful comments. According to the results reported in literature (10.1038/s41467-020-15944-z), it is indeed true that QD thin films are charged by electrons. The charged QDs exhibit a reduced η_r , which is reflected by the lower PL intensity. As we stated in our manuscript, **a major advantage of simultaneous EL-PL measurement is that we can access the η_r in situ when the QDs are electrically pumped.** Therefore, the PL results in our EL-PL measurement actually reflect the precise η_r of the charged QD film.

Comment #4: *The manuscript presents a counter-intuitive scenario where electron leakage via certain paths decreases with increasing voltage. This requires a more detailed explanation, particularly on pages 8, 13, and in Figure 2g.*

Response #4: Thank you for your question. In fact, the number of leakage electrons increase with the driving voltage. However, the leakage channel changes dynamically. The schematic emission spectra shown in Figure 2g illustrate **the relative change of the number of electrons** transport through path 4/5 and to the path 2 (Figure R13). At $V = V_{ELMAX} = (2.10-2.50)$ V, the applied voltage is larger than the bandgap voltage (1.97 V) of the QDs, and thus a considerable amount of holes could inject into QDs, resulting in a maximum γ and EQE. **At this stage, the majority of electrons recombine via path 2, while those overflow via path 4 are reduced and those recombine via path 5 are eliminated, resulting in a very strong QD emission; therefore, compared to the strong QD emission, the emission due to the leakage electrons is relatively weak, as schematically shown in Figure 2g.**

Figure R13 (recapped from Figure 2 of the revised manuscript). Electron transport paths in QLEDs. **f-h** The schematic emission spectra when electrons transport via paths 2, 4, and 5 at $V =$

V_T , V_{ELMAX} and $\gg V_{ELMAX}$, respectively.

Comment #5: *The attribution of the 500 nm EL peak to TFB emission lacks conviction. Literature suggests that TFB's EL should peak between 425-440 nm, which is in line with the PL and EL provided by the authors in Fig. S4 and 5. Typically, the broad emission spectrum of organic molecules should arise from the same excited state and thus emerge simultaneously. If the emissions at 430-460 nm and 460-500 nm are indeed distinct and separable excited species, then the EL (PL) properties of a TFB film should exhibit variance contingent upon the voltage (energy of the incident photons).*

Response #5: Thank you for your insightful comments. The EL at 425-440 nm is the main emission peak of TFB's fluorescence. In fact, it is reported (Figure R14, reprinted from ref. 1) that there is a clear peak at 480-520 nm in the TFB spectrum, which has also been observed in our results (Figure R15, recapped from Supplementary Figure 7 of the revised manuscript). The PL spectrum of the TFB (Figure R16, recapped from Supplementary Figure 6 of the revised manuscript) further confirms that there are multiple emission peaks in TFB.

Although the TFB has multiple PL emission peaks, its EL resulted from the leakage electrons only exhibits the 490-500 nm emission. Considering the disorder of TFB, its LUMO actually consists of multiple energy levels. Most of the leakage electrons can only overflow to the lower LUMO levels of TFB, thus resulting in a long wavelength emission of 490-500 nm. In other words, the energy of electrical excitation caused by leakage electrons is much lower than that of optical excitation, as also discussed in ref. 1, thus resulting in a longer wavelength EL emission.

[Redacted]

Figure R14 (reprinted from ref. 1). EL spectra of a red QLED (Device structure: glass/ITO/PEDOT:PSS/TFB/Red-QDs/bilayer-ZnO/Ag) before electrical aging and after aging at 50.0 mA cm⁻² for 100 h.

Figure R15 (recapped from Supplementary Figure 7 of the revised manuscript). The EL spectrum of device. Device structure: glass/ITO/PEDOT:PSS/TFB/ZnMgO/Al.

Figure R16 (recapped from Supplementary Figure 6 of the revised manuscript). The PL emission spectra of all functional layers (PEDOT:PSS, TFB, ZnMgO) and the red, green, and blue QDs.

To clarify this point, a few sentences have been changed in the revised manuscript. These are as follows.

(Page 12 in the revised Supplementary Information):

Although the TFB has multiple PL emission peaks, its EL resulted from the leakage electrons only exhibits the 490-500 nm emission. Considering the disorder of TFB, its LUMO actually consists of multiple energy levels. Most of the leakage electrons can only overflow to the lower LUMO levels of TFB, thus resulting in a long wavelength emission of 490-500 nm. In other words, the energy of electrical excitation caused by leakage electrons is much lower than that of optical excitation, thus resulting in a longer wavelength EL emission.

References:

- [1] Ye, Y. X. et al. Design of the hole-injection/hole-transport interfaces for stable quantum-dot light-emitting diodes. *J. Phys. Chem. Lett.* **11**, 4649–4654 (2020).

Comment #6: *The manuscript ascribes the red emission in blue QLEDs to interfacial recombination. It is unclear if this concept extends to red QLEDs. If not, an*

explanation is needed. If it does, the analysis might be confounded by the omission of interfacial emission analysis. If the radiative interfacial recombination picture is real, the interfacial emission energy should have a direct relationship with the HOMO level of HTL or the LUMO level of QDs. Confusingly, in literature (reported by the authors, 10.1002/jsid.681), when using TFB or PVK as HTL, the 'radiative interfacial recombination' shows up at the same wavelength.

Response #6: Thank you for your insightful comments. As we stated in the manuscript, the interfacial red emission is only observed in the blue QLEDs. This is because the hole injection in blue QLEDs is the most difficult, and as a result, most holes are accumulated at TFB/blue-QDs interface, thereby resulting in a strong interfacial emission. **Compared to the blue QLEDs, the number of hole accumulation in TFB/red or green-QDs is much lower, thus greatly reducing the interfacial emission. Moreover, even if there is a weak interfacial emission, it is difficult to distinguish such a weak interfacial emission from the strong emission of red or green QDs, as its spectra usually overlap with those of red or green QDs.** Due to the weak emission and the overlapped spectra, we were unable to detect the interfacial emission in red or green QLEDs, but there is still a certain probability that it may be present in red or green QLEDs.

Furthermore, the most important criterion for determining the TFB/QD interfacial recombination is that it tends to weaken with increasing hole injection (by increasing driving voltage). As we stated in manuscript: **“Such an interfacial emission can only occur when there are many carriers accumulated at the interfaces.** At a sub-bandgap voltage of 1.95–2.60 V, hole injection into blue QDs is difficult, and thus a large amount of holes have to accumulate at the TFB/QD interface, thereby triggering the interfacial emission. **By increasing the voltage, more holes can be injected into QDs, and thus, both interfacial and TFB emission are reduced.”**

With regard to our previous report (10.1002/jsid.681), our explanation is as follows: In principle, TFB and PVK have different HOMO levels, QLEDs with different HTLs should indeed exhibit different interfacial emission peaks if the QD is the same. However, we did not observe this phenomenon, and the reason may be that PVK molecules have a higher degree of energetic disorder than TFB, and as a result, the HOMO levels of PVK are broadened (please refer to Figure R17 and ref. 1 for this viewpoint). At the interface between crystalline inorganic QDs and amorphous organic HTLs, the energetic disorder of HTLs leads to more complex interfacial states.

[Redacted]

Figure R17 (reprinted from ref. 1). Theoretically optimized geometries and reorganization energies (λ) for TFB (left) and PF8Cz (right) dimers. Middle, a comparison of the electronic structures.

However, it should be noted that after replacing PVK with P3HT, there is a significant difference in the peak position of the interfacial emission [(Figure R18 (a) and (c))], and the significant redshift of the emission peak is resulted from the higher HOMO level of P3HT [P3HT @ (-4.8eV) vs TFB @ (-5.3 eV)]. This proves that the HOMO change of HTL causes a change in the wavelength of the interfacial emission, further supporting the existence of interfacial recombination.

In addition, there is a significant difference in the peak position of the interfacial emission reported previously compared to that in this manuscript, which is due to the different blue QDs that we used.

[Redacted]

Figure R18 (reprinted from ref. 2). **(a)** Normalized EL spectra of blue quantum dot light-emitting diodes. **(b)** Scheme of exciplex formation. EL, electroluminescence; HTL, hole transport layer. **(c)** Normalized EL spectra of the QLEDs with PVK:25%P3HT.

References:

- [1] Deng, Y. Z. et al. Solution-processed green and blue quantum-dot light-emitting diodes with eliminated charge leakage. *Nat. Photonics* **16**, 505–511 (2022).
- [2] Huang, X. Y. et al. The influence of the hole transport layers on the performance of blue and color tunable quantum dot light-emitting diodes. *J. Soc. Inf. Display* **26**, 470-476 (2018).

Comment #7: *The manuscript omits discussion of the ohmic current, which, according to the data (Fig. 5e and f) and literature (10.1007/s12274-021-3354-7 by the authors or 10.1021/acsnano.9b03507 by another group), significantly influences peak efficiency. The authors should address this aspect.*

Response #7: Thank you for your question. Perhaps our understanding of ohmic current is slightly different.

When $0 < V < V_T$, the electrons can only transport through the inter-bandgap levels of all functional layers, leading to the generation of ohmic current or inter-band leakage current. The ohmic current (the Ohmic region in Figure R26) is present during the entire operation period, but its impact on device efficiency is negligible,

since its value is order of magnitude smaller than that of the recombination current.

In Figure 5e and Figure 5f, we hold the opinion that the current drop is due to the reduction of leakage electrons, which should not be considered as ohmic current. Due to the reduction of leakage current (not ohmic current!), the optimized QLEDs exhibit a higher EQE, as shown in Figure 5e and 5f.

Comment #8: *On page 2 of the main text (line 12), and page 4 of the supplementary information (line 4), "no all" should be corrected to "not all".*

Response #8: Thank you very much for your careful review. We have carefully reviewed all errors and made necessary corrections.

Comment #9: *As pointed out by the authors, the turn-on voltage of QLEDs is influenced by the noise-floor of detection system and the device temperature, which thus lacks of a physical entity. As a result, the discussion on different transport paths classified by driven voltage below or above turn-on voltage is ambiguous (Page 7).*

Response #9: Thank you for your insightful comments. Although the turn on voltage is affected by both noise-floor of detection system and the ambient temperature, it usually is defined as the voltage corresponding to a luminance of 1 nit at room temperature. With this definition, the discussion on different transport paths classified by driven voltage below or above turn-on voltage is clear.

Comment #10: *The origin of the dark spot shown in Figure 1f should be explained to avoid any confusion regarding the experimental setup or results.*

Response #10: Thank you for your suggestion. The photographs in Figure 1f were taken through the lens of PR670 spectrometer. **The dark spot is not originated from the device; it indicates the size of the aperture o the PR670, which determines the light input and brightness detection range of PR670.**

To clarify this point, a few sentences had been added to the revised manuscript, as follows.

(Page 7-8 in the revised manuscript):

The photographs were taken through the lens of a PR670 spectrometer. The dark spot is not originated from the QLEDs; it is the indication of the aperture in the PR670.

Comment #11: *The manuscript should include details on the calibration process for the single-point detector system used for acquiring broad-band spectra. The detection efficiency of detector and diffraction efficiency of grating of monochromator may changes significantly against wavelength. The omission of this information is critical*

as it affects the accuracy and reliability of the data presented.

Response #11: Thank you for your helpful suggestion. To clarify this point, Supplementary Note 3 has been revised as follows.

(Page 6 in the revised Supplementary Information):

Supplementary Note 3. Tracing the leakage electrons in QLEDs via single photon counting (SPC) technique.

The schematic measurement setup is shown in Figure 3a in the main text. The SPC technique utilizes the inherent discrete nature of photon detector output signals under weak light illumination. By employing pulse discrimination and digital counting techniques, it allows for the identification and extraction of extremely faint light signals. In this study, the SPC integrated within the Edinburgh FS5 system was utilized, which is commonly employed for fluorescence lifetime measurements. We employed this system to detect the weak light signals generated within the QLED due to electron leakage. The setup also incorporates a grating monochromator that enables the selective detection of the photons at a specific wavelength.

During the testing process, the QLED sample was placed at the receiving end of the single photon counter, and the testing area was sealed to isolate ambient light. Subsequently, by driving the QLED through a source meter, the number of photons was recorded for each spectrum at different driving voltages.

Other details:

- (1) The grating monochromator integrated in the single photon counter enables the selective detection of the photons at a specific wavelength. The split signal goes directly to the detector without additional amplification.
- (2) The detector in the single photon counter is calibrated with a broad-spectrum standard light source, ensuring an accurate and error-free response at visible wavelength.
- (3) The kinetic scan mode with a corresponding bandwidth was used during the test. The photon counts for each data point were averaged over a collection time of 30 s. During the testing, the emission of the QLED was very stable.
- (4) The ratio in Figure 3e is mainly calculated based on the EL results detected by single photon counter.
- (5) The device position was fixed during testing.

Comment: Overall, this work could become an interesting publication if the authors could clarify the above issues. Alternatively, the authors might eliminate those repetitive studies/analysis and concentrate on the so-called “interfacial recombination” (Point 6 above). To our knowledge, this could be potentially a new and important finding. In this regard, further experimental (such as with different

types of quantum dots with variable LUMO levels) and theoretical (such as calculations of possible rates of this unique charge transport channel) studies are expected to solidify their claims.

Our Response: We would like to thank you once again for your efforts in reviewing this manuscript. We greatly appreciate your insightful comments.

In this work, we have used the SPC technique to examine the interfacial recombination in QLEDs and provide new quantitative results. In future research, we will explore whether there are deeper new insights based on your valuable comment 6. We would be happy to discuss this in more detail at that time.

We hope our responses/revisions satisfactorily address all your concerns. Once again, we thank you for your constructive and helpful suggestions!

 **Response to Reviewer #5 (co-reviewed with Reviewer #4)**

General comment: *I co-reviewed this manuscript with one of the reviewers who provided the listed reports. This is part of the Nature Communications initiative to facilitate training in peer review and to provide appropriate recognition for Early Career Researchers who co-review manuscripts.*

Our response: We would like to express our gratitude for your efforts in reviewing this manuscript. We greatly appreciate your insightful comments on our manuscript and your constructive suggestions are invaluable in helping us to improve the quality of this paper.

REVIEWER COMMENTS

Reviewer #1 (Remarks to the Author):

The revision appears fully satisfactory for the publication.

Reviewer #2 (Remarks to the Author):

The authors addressed all issues and comments raised by the reviewers and provided appropriate answers. And the manuscript was revised accordingly. Therefore, this reviewer recommends its acceptance for publication.

Reviewer #3 (Remarks to the Author):

The authors have carefully answered all the questions raised by the reviewers. I am satisfied with the responses to my questions.

The revised manuscript would be published as is.

Reviewer #4 (Remarks to the Author):

Please see my comments in attached .doc file.

Reviewer #5 (Remarks to the Author):

In the revised manuscript, the authors have made considerable strides in responding to the feedback from the previous review cycle. Nonetheless, certain critical issues demand further elucidation, as detailed in the subsequent points.

The authors have underscored the innovative application of the SPC technique in this study, which has facilitated the tracking of electron transport pathways. This has led to the identification of a faint photon signal that is commonly undetected by traditional detection methods. Despite the noteworthy detection of this elusive signal, the precise source of the signal, which is pivotal to the study's findings, has not been sufficiently clarified by the authors. This unresolved issue stands as the primary concern that warrants the authors' attention.

1. In the manuscript, the authors propose that 'path 4' is responsible for the TFB emission, with its most compelling evidence being the emission's peak at 490 nm. The lack of TFB emission within the 425-440 nm range is attributed to inadequate electron energy from electrical excitation. To reinforce this assertion, it would be beneficial to **observe a gradual increase in higher energy peaks as the bias voltage is incremented, which could be demonstrated in either the Red-QLED or TFB-emitter device.** Furthermore, **the photoluminescence (PL) spectrum should exhibit discernible changes when the excitation photon energy is varied.** These points, previously noted in comment #5, are critical for validating the proposed mechanism.
2. The proposed 'path 3', leading to interfacial emission at the HTL/QD interface, finds substantial support in the referenced study (10.1002/jsid.681). A key piece of evidence supporting this claim is the observed red-shifted 'interfacial emission' in the structure of ..P3HT/QD/ETL/..(compared with TFB-based QLEDs), which is indicative of the emission process occurring at the HTL/QD interface. To strengthen the manuscript's scientific rigor and clarity, it is recommended that **this critical evidence be incorporated into the revised manuscript.**
3. In the manuscript, the authors emphasize the role of 'Ohmic current' and its negligible impact on efficiency when the bias is below the threshold voltage (V_t). However, **an estimation of the contributions of ohmic current and leakage current is notably absent** (I cannot catch Figure R26 in the response letter), which is crucial for understanding their effects on the EQE in regions of low current density. In my opinion, the term 'leakage current' is a component of the 'ohmic current', provided that the current shows a linear dependence on voltage, as illustrated in Figure 2e. It is reasonable to hypothesize that a decrease in current below V_t could result in a higher peak EQE. However, **the origin of this current is suggested to be the leakage hole current into the ETL,** as supported by the referenced literature (10.1021/acsnano.9b03507). Furthermore, in authors' previous publication, this **'leakage current' below V_t is also attributed to leakage current at QD/ETL interface** (10.1007/s12274-020-3091-3), which is observed in the response letter (Fig. R3) as well. It is highly possible that hole leakage current could be mitigated with increased QD packing density or the incorporation of a blocking layer. As a result, the relationship between decrease of 'leakage current' and decrease of interfacial recombination at HTL/QD interface lacks convincing evidence.

[Redacted]

Reprinted from 10.1007/s12274-020-3091-3

✚ Response to Reviewer #1 & #2 & #3

General comment:

#1: The revision appears fully satisfactory for the publication.

#2: The authors addressed all issues and comments raised by the reviewers and provided appropriate answers. And the manuscript was revised accordingly. Therefore, this reviewer recommends its acceptance for publication.

#3: The authors have carefully answered all the questions raised by the reviewers. I am satisfied with the responses to my questions. The revised manuscript would be published as is.

Our response: Thank you very much for your efforts in reviewing our work. We sincerely appreciate your constructive comments, which will certainly help to improve the quality of this paper.

✚ Response to Reviewer #4 & #5

General comment: *In the revised manuscript, the authors have made considerable strides in responding to the feedback from the previous review cycle. Nonetheless, certain critical issues demand further elucidation, as detailed in the subsequent points.*

The authors have underscored the innovative application of the SPC technique in this study, which has facilitated the tracking of electron transport pathways. This has led to the identification of a faint photon signal that is commonly undetected by traditional detection methods. Despite the noteworthy detection of this elusive signal, the precise source of the signal, which is pivotal to the study's findings, has not been sufficiently clarified by the authors. This unresolved issue stands as the primary concern that warrants the authors' attention.

Our response: Thank you very much for your efforts in reviewing our work. We sincerely appreciate your constructive comments, which will certainly help to improve the quality of this paper. We will continue to discuss and address your concerns point-by-point in the following responses.

Comment #1: *In the manuscript, the authors propose that 'path 4' is responsible for the TFB emission, with its most compelling evidence being the emission's peak at 490 nm. The lack of TFB emission within the 425-440 nm range is attributed to inadequate electron energy from electrical excitation. To reinforce this assertion, it would be beneficial to **observe a gradual increase in higher energy peaks as the bias voltage is incremented, which could be demonstrated in either the Red-QLED or TFB-emitter device.** Furthermore, **the photoluminescence (PL) spectrum should exhibit discernible changes when the excitation photon energy is varied.** These*

points, previously noted in comment #5, are critical for validating the proposed mechanism.

Response #1: Thank you for your insightful comments. Indeed, the TFB emission of 490 nm is the most direct evidence for ‘path 4’. To further confirm this point, as you suggested, we examined the TFB emission in the red QLED at different driving currents. As shown in Figure R1 (recapped from Supplementary Figure 8 of the revised manuscript), the EL spectra of the red QLED showed a gradual increase in the 420-450 nm region as the driving current was increased, proving that the increase in the number of leakage electrons could more fully excite the fluorescence emission of TFB. This suggests that the increase in electrical pumping can indeed cause the TFB emission to shift towards shorter wavelengths. However, it should be noted that since the percentage of leakage electrons in the device is small, the TFB emission is still mainly concentrated around 490 nm.

Figure R1 (recapped from Supplementary Figure 8 of the revised manuscript). TFB emission in the red QLED at different driving currents.

In addition, we are unable to provide similar changes in PL. PL is an energy down-conversion process. For the three PL peaks in the 420-510 nm range in TFB, it is difficult to achieve adequate independent excitation by varying the excitation photon energies. This is because the absorption of TFB is mainly concentrated before 425 nm [ref. 1, ref. 2], as shown in Figure R2. Therefore, TFB can only effectively absorb the excitation photon energy higher than (1240/425) eV. The PL excitation scan of TFB film (Figure R3) also confirms this point.

Furthermore, if the photon energy (for example Ex.=360 nm) is kept constant and only the excitation power is varied, the entire TFB emission from 420-510 nm is always excited, only the intensity is varied and the peak positions are consistent.

We believe that the TFB emission at the TFB/QD interface caused by electron leakage

is much more complex than in PL. So far, the explanations and evidence we have provided are reasonable. But we will continue to focus on this to see if there are any new discoveries.

[Redacted]

Figure R2 (reprinted from ref. 1, ref. 2). (a) Absorption spectra of spin-coated TFB films before (blue lines) and after (red lines) rinsing of the QD ink [ref. 1]. (b) Absorption and photoluminescence spectra for TFB [ref. 2].

Figure R3 PL excitation scan of TFB film.

Figure R1 has been added as Supplementary Figure 8 to the revised Supplementary Information:

Supplementary Figure 8. Confirmation of TFB fluorescent emission localization.

The lack of TFB emission in the 420-450 nm range indicates the insufficient EL excitation of TFB. To further confirm this point, we examined the TFB emission in the red QLED at different driving currents. The EL spectra showed a gradual increase in the 420-450 nm region as the driving current was increased, proving that the increase in the number of leakage electrons could more fully excite the fluorescence

emission of TFB. However, it should be noted that since the percentage of leakage electrons in the device is small, the TFB emission is still mainly concentrated around 490 nm.

Reference:

- [1] Tang, H. et al. Improved Ink-Jet-Printed CdSe Quantum Dot Light-Emitting Diodes with Minimized Hole Transport Layer Erosion. *ACS Applied Electronic Materials* **3**, 3005–3014 (2021).
- [2] Renzi, W. et al. Exploring the Experimental Photoluminescence, Raman and Infrared Responses and Density Functional Theory Results for TFB Polymer. *Synthetic Metals* **236**, 24–30 (2018).

Comment #2: *The proposed ‘path 3’, leading to interfacial emission at the HTL/QD interface, finds substantial support in the referenced study (10.1002/jsid.681). A key piece of evidence supporting this claim is the observed red-shifted ‘interfacial emission’ in the structure of ../P3HT/QD/ETL/.. (compared with TFB-based QLEDs), which is indicative of the emission process occurring at the HTL/QD interface. To strengthen the manuscript’s scientific rigor and clarity, it is recommended that **this critical evidence be incorporated into the revised manuscript.***

Response #2: Thank you for your suggestion. To strengthen and clarify this point, our previous results were reprinted and have been added as Supplementary Figure 13 (Figure R4 below) to the revised Supplementary Information.

Figure R4 (reprinted from Supplementary Figure 13 of the revised manuscript). Confirmation of the interfacial recombination.

Supplementary Figure 13. Confirmation of the interfacial recombination. To further demonstrate the interfacial recombination, we reprinted the interfacial emission results from our previous work [7]. In principle, QLEDs with different HTLs should exhibit different interfacial emission peaks if the QD is the same. After replacing TFB with P3HT, there is a significant difference in the peak position of the

interfacial emission and the significant redshift of the emission peak, which results from the higher HOMO level of P3HT [P3HT @ (-4.8 eV) vs TFB @ (-5.3 eV)]. This proves that the HOMO change of HTL causes a change in the energy of the interfacial emission, further supporting the existence of interfacial recombination.

In addition, there is a significant difference in the peak position of the interfacial emission reported previously compared to that in this work, which is due to the different blue QDs that we used.

Comment #3: *In the manuscript, the authors emphasize the role of ‘Ohmic current’ and its negligible impact on efficiency when the bias is below the threshold voltage (V_t). However, an estimation of the contributions of ohmic current and leakage current is notably absent (I cannot catch Figure R26 in the response letter), which is crucial for understanding their effects on the EQE in regions of low current density. In my opinion, the term ‘leakage current’ is a component of the ‘ohmic current’, provided that the current shows a linear dependence on voltage, as illustrated in Figure 2e. It is reasonable to hypothesize that a decrease in current below V_t could result in a higher peak EQE. However, the origin of this current is suggested to be the leakage hole current into the ETL, as supported by the referenced literature (10.1021/acsnano.9b03507).*

Furthermore, in authors’ previous publication, this ‘leakage current’ below V_t is also attributed to leakage current at QD/ETL interface (10.1007/s12274-020-3091-3), which is observed in the response letter (Fig. R3) as well. It is highly possible that hole leakage current could be mitigated with increased QD packing density or the incorporation of a blocking layer. As a result, the relationship between decrease of ‘leakage current’ and decrease of interfacial recombination at HTL/QD interface lacks convincing evidence.

Response #3: Thank you for your insightful comments. We hope to address your concerns as best we can in the discussion that follows.

Strictly speaking, ohmic current is a part of leakage current, but not all leakage current can be regarded as ohmic current. In this manuscript we have extracted ohmic currents for separate discussion, mainly because ohmic currents, as considered here, have a negligible effect on efficiency.

All currents in a QLED system affect the EQE, but the reasons for the sentence “The ohmic current is present during the entire operation period, but its impact on device efficiency is negligible, ...” we stated are as follows.

- (1) We need to discuss Figure 2e in detail. Figure R5a shows the fitting results of the J-V characteristics of a red QLED. We can see that the current is divided into two intervals: ① Ohmic current region, where $J \propto V$. ② Diode recombination

current and leakage current. In the ohmic current region, the electrons only transport through the inter-bandgap levels of all functional layers. **At this point, the electrons forming the ohmic current do not pass through any Schottky interface, which is very similar to an ohmic contact, so the J-V characteristic is linear.**

- (2) The ohmic current (the green dash line in Figure R5b is the ‘hypothetical ohmic current’) is present during the entire operation period, its value is order of magnitude smaller than that of the recombination current. For example, when $V=2.0$ V, the ohmic current is about $6 \cdot 10^{-3}$ mA cm⁻², which is three orders of magnitude smaller than the total current of $3 \cdot 10^0$ mA cm⁻². When $V=5.0$ V, the ohmic current is about $1.5 \cdot 10^{-2}$ mA cm⁻², which is over four orders of magnitude smaller than the total current of $8 \cdot 10^2$ mA cm⁻². Even when $V=1.6$ V (\approx turn-on voltage), the ohmic current ($4 \cdot 10^{-3}$ mA cm⁻²) is still one order of magnitude smaller than the total current ($3 \cdot 10^{-2}$ mA cm⁻²). **As a result, the ohmic current is always less than 10% of the total current, and less than 0.01% over most of the current range. Therefore, we stated that its impact on device efficiency is negligible.**

Figure R5 a Log-log plot of current density-voltage characteristics of a red QLED. b ‘hypothetical ohmic current’ (green dash line) during the entire operation period.

- (3) **When the J-V curve becomes non-linear below the turn-on voltage, it indicates that (2) cannot be treated as an ohmic region.** The leakage currents, such as the leakage hole current into the ETL (10.1021/acsnano.9b03507), the leakage current at QD/ETL interface, significantly affect the efficiency (10.1007/s12274-020-3091-3), as stated in the above literature. At this point, the leakage electrons should pass through the Schottky interface to form the leakage current, and its J-V characteristic is no longer linear. It can no longer be treated as an ohmic current. At this time, the leakage current has a significant effect on the

efficiency. This is also shown in Figures 5e and 5f of the revised manuscript. **Therefore, your statement “It is reasonable to hypothesize that a decrease in current below V_t could result in a higher peak EQE.” is absolutely correct when the leakage current refers to the (2).**

Therefore, in order to clarify this point, a few sentences have been changed in the revised manuscript. These are as follows.

(Page 8 in the revised manuscript):

It is important to note that when the J-V curve becomes non-linear in this voltage range, it can no longer be defined as an ohmic current. At this point, variations in this current can significantly affect the efficiency of the QLEDs²⁹.

Regarding your concern that “the relationship between decrease of ‘leakage current’ and decrease of interfacial recombination at HTL/QD interface lacks convincing evidence.”, the discussion is as follows:

The reduced leakage current is highly correlated with the reduced emission of interfacial recombination. To prove the above statement, we have reduced the leakage current in blue QLEDs by replacing the TFB HTL with PF8Cz HTL. The use of PF8Cz HTL can effectively reduce the leakage current in blue QLEDs due to its low electron affinity and reduced energetic disorder (Figure R6, reprinted from ref. 1), as reported by Deng et. al. [ref. 1]. As shown in Figure R7, with PF8Cz, the resulting QLED exhibits a reduced leakage current and a higher EQE, a similar result with that in Figures 5e and 5f. Figure R8 shows the EL spectra obtained by SPC. **It can be observed that the device with PF8Cz exhibits a weaker interfacial emission (marked with red circle), which confirms the reduction of electron leakage.**

[Redacted]

Figure R6 (reprinted from ref. 1). Theoretically optimized geometries and reorganization energies (λ) for TFB (left) and PF8Cz (right) dimers. Middle, a comparison of the electronic structures.

Figure R7. The J-V and EQE comparisons of blue QLEDs using TFB and PF8Cz HTL respectively.

Figure R8 (reprinted from Supplementary Figure 12 of the revised manuscript). The spectra of blue QLEDs were obtained using SPC under different driving voltages.

Reference:

- [1] Deng, Y. Z. et al. Solution-Processed Green and Blue Quantum-Dot Light-Emitting Diodes with Eliminated Charge Leakage. *Nat. Photonics* **16**, 505–511 (2022).

REVIEWERS' COMMENTS

Reviewer #4 (Remarks to the Author):

The authors have fully satisfied the concerns I had raised. Upon reviewing the revisions, I am confident that the manuscript is now ready for publication in its current form.

Reviewer #5 (Remarks to the Author):
